# Muscarinic modulation of M and h currents in gerbil spherical bushy cells

**Charlène Gillet[1], Stefanie Kurth[2], Thomas Kuenzel [1,2] ***

**1** Auditory Neurophysiology Group, Department of Chemosensation, RWTH Aachen University, Worringerweg, Aachen, Germany, **2** Department of Chemosensation, RWTH Aachen University, Worringerweg, Aachen, Germany

\* kuenzel@bio2.rwth-aachen.de

## Abstract

Descending cholinergic fibers innervate the cochlear nucleus. Spherical bushy cells, principal neurons of the anterior part of the ventral cochlear nucleus, are depolarized by cholinergic agonists on two different time scales. A fast and transient response is mediated by alpha-7 homomeric nicotinic receptors while a slow and long-lasting response is mediated by muscarinic receptors. Spherical bushy cells were shown to express M3 receptors, but the receptor subtypes involved in the slow muscarinic response were not physiologically identified yet. Whole-cell patch clamp recordings combined with pharmacology and immunohistochemistry were performed to identify the muscarinic receptor subtypes and the effector currents involved. Spherical bushy cells also expressed both M1 and M2 receptors. The M1 signal was stronger and mainly somatic while the M2 signal was localized in the neuropil and on the soma of bushy cells. Physiologically, the M-current was observed for the gerbil spherical bushy cells and was inhibited by oxotremorine-M application. Surprisingly, long application of carbachol showed only a transient depolarization. Even though no muscarinic depolarization could be detected, the input resistance increased suggesting a decrease in the cell conductance that matched with the closure of M-channels. The hyperpolarization-activated currents were also affected by muscarinic activation and counteracted the effect of the inactivation of M-current on the membrane potential. We hypothesize that this double muscarinic action might allow adaptation of effects during long durations of cholinergic activation.

## Introduction

Cholinergic top-down connections exist at all levels of the auditory pathway to modulate sound information processing [1]. A well-documented descending system is the olivo-cochlear projection to the inner ear. Here, cholinergic activation reduces the excitability of the inner ear and facilitates unmasking of sounds in a noisy background by inhibiting the constant response to the noise and thus reducing wide-spread adaptation [2]. The olivo-cochlear bundle also sends collaterals into the cochlear nucleus [2,3–8]. In addition, about one quarter of cholinergic connections to the ventral cochlear nucleus comes from the pontomesencephalic

**Data Availability Statement:** Data are available in the following repository: https://doi.org/10.12751/g-node.4bdb22.

**Funding:** This work was funded by the DFG Priority Program 1608 "Ultrafast and Temporally Precise Information Processing: Normal and Dysfunctional

Hearing" with German Research Foundation Grants KU2529/2-1 (TK), KU2529/2-2 (CG and TK) and by German Research Foundation regular grant KU2529/3-1 (TK). The funders had no role in study design, data collection and analysis, decision to publish, or preparation of the manuscript.

**Competing interests:** The authors have declared that no competing interests exist.

tegmentum [8]. The function of these descending connections into the ventral cochlear nucleus (VCN) are not well understood. In the VCN, acetylcholine acts on stellate [9] and spherical bushy cells (SBC) [10] through both nicotinic and muscarinic receptors. The SBC resting membrane potential depolarized on two different time scales following a brief application of carbachol, a broad cholinergic agonist. Nicotinic α7 receptors mediate a fast, transient depolarization while muscarinic receptors cause a slow, long-lasting depolarization. Although immunohistochemical data show the presence of M3 muscarinic receptors on SBC bodies [5], the presence of other muscarinic subtypes and the currents involved in muscarinic depolarization remain unresolved. We hypothesize, that muscarinic modulation could act through slow voltage-activated currents, like the M- and h-currents, in spherical bushy cells.

The M-current ($I_M$) is a slowly activating, non-inactivating voltage-gated potassium outward current which is sensitive to muscarine [11]. Muscarinic agonists induce a slow depolarization [11–13] by inhibiting $I_M$ and this effect is usually accompanied by a decrease in membrane conductance. The voltage-gated potassium channel Kv7 underlies $I_M$ [14] but there is evidence that other channels, like two-pore potassium channels, can also generate $I_M$ [15]. To our knowledge, the presence of $I_M$ in gerbil SBCs has not yet been demonstrated.

The hyperpolarization-activated cation current or $I_h$, is present in brainstem auditory neurons including SBCs [16]. It can affect the resting membrane potential and influence excitability [17–19]. $I_h$ is mediated by hyperpolarization-activated cyclic nucleotide-gated (HCN) channels [20]. In vitro data have shown that $I_h$ is sensitive to muscarinic activation in striatal cholinergic interneurons [21].

In this study, we identified the muscarinic receptor subtypes responsible for the SBC response to the application of cholinergic agents, and their associated currents. For that, whole-cell patch clamp recordings combined with pharmacology and immunohistochemistry were performed. We demonstrated that SBCs expressed $I_M$ and that both $I_M$ and $I_h$ were affected by the application of oxotremorine-M (oxoM). Additionally, our immunohistochemical data showed that SBCs expressed two further subtypes of muscarinic receptor, M1 and M2.

## Materials and methods

### Animals

All experiments were performed in the Institute for Biology 2, RWTH Aachen University, Aachen, Germany in accordance with the European Communities Council Directive of 24 November 1986 (86/609/EEC) and authorized by local state authorities (North Rhine-Westphalia State Agency for Nature, Environment and Consumer Protection, Recklinghausen, Germany). A total of 46 gerbils (Meriones unguiculatus) of either sex aged from post-natal day (P) 14 to 30 were used.

### Slice preparation

The animals were deeply anesthetized with isoflurane and quickly decapitated. The brain was removed from the skull and the brainstem was prepared in ice-cold cutting buffer for slicing. The cutting buffer contained (in mM): 215 sucrose, 10 glucose, 2.5 KCl, 4 $MgCl_2$*$6H_2O$, 0.1 $CaCl_2$*$2H_2O$, 1.25 $NaH_2PO_4$*$2H_2O$, 25 $NaHCO_3$, 3 myo-inositol ($C_6H_{12}O_6$), 2 sodium pyruvate ($C_3H_3NaO_3$), 0.5 L-ascorbic acid ($C_6H_8O_6$), and was bubbled with 95% $O_2$ and 5% $CO_2$ to a pH value of 7.4 (308 mOsmol). The brainstem was cut either in frontal or parasagittal 180–250 μm slices by a vibrating microtome (VT1200S, Leica Biosystems, Nussloch, Germany). Then the slices were incubated for at least one hour in artificial cerebrospinal fluid (ACSF) at room temperature (RT, 25°C) containing (in mM): 125 NaCl, 2.5 KCl, 1 $MgCl_2$, 2 $CaCl_2$, 1.25 $NaH_2PO_4$, 25 $NaHCO_3$, 10 glucose, 3 myo-inositol ($C_6H_{12}O_6$), 2 sodium pyruvate

($C_3H_3NaO_3$), 0.5 L-ascorbic acid ($C_6H_8O_6$), bubbled with 95% $O_2$ and 5% $CO_2$ to a pH value of 7.4 (314 mOsmol).

## Electrophysiology

The slices containing the rostral part of the AVCN were placed in the recording chamber and perfused with oxygenated ACSF (100 ml/h) at RT under a fixed-stage microscope with IR-DIC and fluorescent imaging (Nikon Eclipse FN-1 microscope equipped with DS-Qi1MC camera and DC-U3 camera controller, Nikon Instruments, Japan). The patch electrodes were made from borosilicate glass filament capillaries (Science Products GmbH) with a horizontal DMZ Universal Puller (Zeitz-Instruments). The pipet resistance was between 2 to 5 MΩ (with over-pressure) when the pipets were filled with a gluconate-based internal solution containing (in mM): 100 K-gluconate, 40 KCl, 0.1 $CaCl_2$, 10 HEPES, 1.1 EGTA, 2 Mg-ATP, 0.4 GTP, 0.1 Alexa-488 hydrazide (Thermo Fischer Scientific), 3 mg/ml biocytin (Thermo Fischer Scientific), adjusted to a pH of 7.2 with 1 M KOH (280 mOsm). The SBCs were recorded in whole-cell configuration with a HEKA EPC10 USB double patch clamp amplifier. The data were acquired by HEKA patchmaster software (HEKA Elektronik Dr. Schulze GmbH, Lambrecht/ Pfalz, Germany). The recordings were low-pass filtered at 2.7 kHz and sampled at 50 kHz (100 kHz for the spontaneous miniature activity). The junction potential was corrected online for $I_M$ and $I_h$ measurements (calculated at -11 mV). The series resistance (Rs) was uncompensated. Only cells with a Rs < 35 MΩ and with less than 30% change during the recordings were used. All electrophysiological recordings were made in slices from gerbils aged between P14-P25. This range was defined to get cell recordings as stable as possible after hearing onset, estimated at P12 [22].

**$I_M$ current measurement.** In voltage-clamp mode, the cells were initially held at -60 mV. The cells were then clamped to -20 mV for 1.8 s and hyperpolarizing steps from -70 to -30 mV with 10 mV increment were applied for 2 s before returning to -20 mV. For each voltage step, $I_M$ amplitude was measured as the difference between the instantaneous current ($I_{inst}$; maximum current in the first 100 ms of the 2 s voltage step) at the beginning of the step, caused by the leak conductance (or very rapidly voltage activated conductances), and the steady state current ($I_{ss}$; average current of the last 150 ms of the 2 s voltage step) at the end. ZD7288 (20 μM; Bio Techne/Tocris UK) was added to ACSF to block $I_h$. In Fig 1, α-Dendrotoxin (αDTX, 50nM, Biozol, Germany) was also added in ACSF to block the Kv1.1 and Kv1.2 channels. Oxo-tremorine-M (oxoM, 1 μM; Bio Techne/Tocris UK) was locally applied with a perfusion pencil

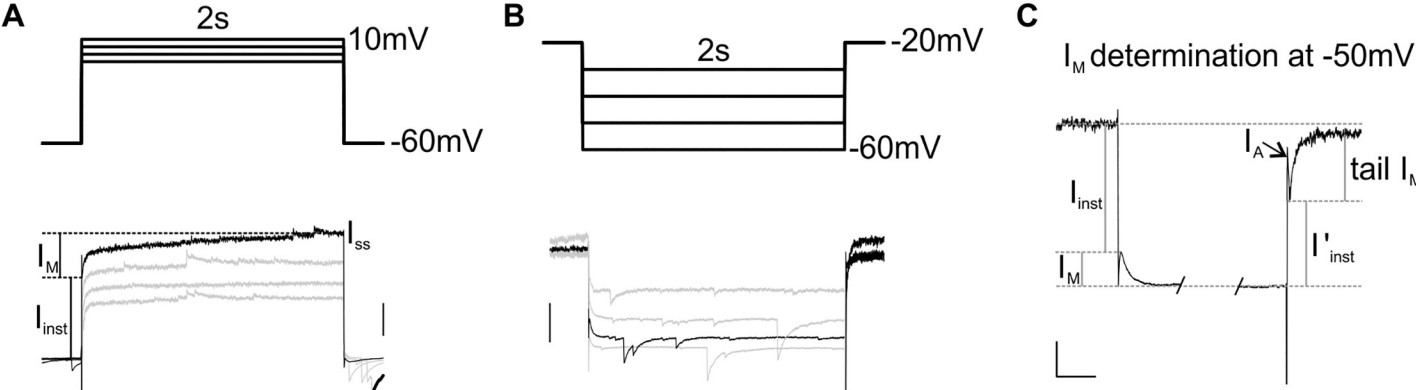

**Fig 1. Presence of $I_M$ in SBCs.** (A) Activation protocol (upper part) and its current response (lower part); scale: 500 pA. (B) Deactivation protocol (upper part) and its current response (lower part); scale: 200 pA. (C) Higher magnification of the current response shown in B at voltage step -50 mV; scale: 100 ms/100 pA.

in absence or after washing in XE991 dihydrochloride (XE991, 10 µM; Bio Techne/Tocris UK).

**$I_h$ current.** In voltage-clamp mode, the cells were held at -60 mV and stepped to voltages from -110 mV to -30 mV with 10 mV increment for 2 s. The current $I_h$ was calculated as the difference between the instantaneous current and the steady-state current. $I_M$ was blocked by adding XE991 (10 µM) to ACSF. OxoM (1 µM) was locally applied in absence or after washing in ZD7288 (20 µM).

**Membrane potential and excitability monitoring.** In current-clamp mode, a set of 10 current steps was injected onto the SBCs every 30 s for 20 minutes. From every set of step currents a current-voltage curve was constructed from which the input resistance (IR) was derived as a linear fit to the subthreshold part of the current-voltage curve. Also, the resting membrane potential (RMP) was taken from the mean of the current clamp recordings prior to stimulus current injection. From a strongly hyperpolarizing current step the amount of "voltage sag" was measured as the difference between the maximum and the steady-state voltage deflection. From a suprathreshold current step the AP threshold potential was determined by thresholding the relation between the change of membrane potential (in mV/ms) and the membrane potential (i.e. a "phaseplane" plot). A threshold rate of change of 50mV/ms was taken as indicative of the onset of the rising phase of the AP. This procedure yields the membrane potential at which the steep incline of the AP begins, which we took as an estimate of the AP threshold. Changes in RMP, IR, voltage sag and AP threshold potential were observed upon wash-in of carbachol (carbamylcholine chloride, 500 µM; Sigma Aldrich Germany), oxotremorine-M (1 µM) or 4DAMP (1 µM; Bio Techne/Tocris UK). The first 2.5 min of these experiments were discarded (duration that the cell needed to stabilize). The values for RMP, IR, voltage sag and AP threshold potential were normalized to the mean of the values obtained during the 2.5 min control condition. For statistical analysis, the mean of the values obtained during 2.5 min per cell for each condition were averaged and compared (i.e., all values for control condition, 2.5 min where the effect was maximal with the agent tested and the last 2.5 min for washout, unless stated otherwise) by one way ANOVA with repeated measures.

**Evoked and spontaneous miniature EPSCs.** An Iso-Flex stimulus isolator (A.M.P.I., Jerusalem, Israel) was connected to a 250 µm bipolar tungsten electrode (Micro Probe Inc., Gaithersburg, USA) of 1.5 MΩ impedance placed in the AVCN near the patched cell to electrically stimulate AN fibers as described in [10]. The stimulus intensity was set to 1.2 times the threshold intensity of the activation of one single input (range: 14–28 V). The stimulation protocol consisted of a stimulus train of 30 stimuli at 50 Hz, 100 Hz or 200 Hz, repeated 10 times with 30 s interval and was performed before and after washing in carbachol (500 µM) for 10 minutes. The EPSC 10%-90% rise time was determined and the EPSC decay time constant was calculated by fitting a single exponential function to the decaying slopes of EPSC. The eEPSC amplitude was determined by the difference between the crossing point of the rising slope and the baseline linear fits, and the maximal deflection point. To quantify short-term depression, the amplitude of eEPSCs for each stimulus was averaged over 10 repetitions and normalized to the first eEPSC amplitude. The size of the readily-releasable pool was estimated with the back-extrapolation method [23] from 100Hz pulse-trains. Using the estimated pool-size in nA and the amplitude of the first EPSC, the initial release probability was calculated. We furthermore estimated the amount of asynchronous release as in [24].

Spontaneous miniature EPSCs (mEPSCs) were recorded for several minutes before, during and after application of carbachol (500 µM). The mEPSCs were detected by a custom-made automated detection algorithm briefly described below. This algorithm is based on an algorithm developed by [25]. First, ten to twenty mEPSCs were detected manually for each cell and averaged to create a template. Second, low-frequency noise was removed by subtracting a

polynomial function (9th-order, unconstrained fitting procedure) fitted to the whole 1000ms current trace. Third, the template was slid along the current trace. To fit the data, the template was optimally scaled and offset at each position of the trace. Finally, a detection criterion was determined by the template scaling and the goodness of the fit at each point of the data. The event was detected when this criterion crossed a threshold set manually for each cell.

All EPSC recordings were made in presence of glycine and GABA receptor blockers: strychnine (1 μM; Sigma Aldrich Germany), gabazine (10 μM; Abcam UK) and CGP 55845 hydrochloride (2 μM; Bio Techne/Tocris UK).

## Biocytin-streptavidin staining

During recordings, the cells were filled with biocytin contained in the internal solution (see above) to morphologically confirm the cell type afterwards. At the end of the experiment, the slices were immersed in 4% paraformaldehyde (PFA) in phosphate buffered saline (PBS) overnight for fixation and transferred in PBS. After rinsing several times with PBS, the slices were washed with 0.1% Triton X-100 in PBS. Then, the slices were incubated for 2.5h at RT in a streptavidin solution containing: 0.1%Triton X-100, 1% bovine serum-albumine (BSA) in PBS, Alexa Fluor® dye streptavidin conjugates (1/800, Cat. No. S11223, Thermo Fisher Scientific, Germany). The slices were washed several times, first with PBS, second with 0.3% Triton X-100 in Tris buffered saline (TBS) and finally with TBS only. At the end, a 4'-6-diamidin-2-phenylindol-dye (DAPI) staining was performed (0.1μg/ml in PB). Each slice was placed on a coverslip inside of a 240 μm thick adhesive frame (Grace Bio-Labs, USA). Fluoprep (bioMérieux, UK) was applied and the slices sealed in with another coverslip placed firmly on the adhesive frame. This technique allowed direct imaging of slices up to 240μm without distortions due to pressure exerted by the cover slip. Neurons in the slices were imaged with a laser-scanning confocal microscope (TCS SP2, Leica microsystems, Germany). The z-resolution in stacks of confocal images was set to 0.5 μm. The contrast and the brightness of all images were adjusted using ImageJ (RRID: SCR_003070).

## Immunohistochemistry

For immunohistochemistry, P30 gerbils (n = 3) were very deeply anesthetized with an intraperitoneal injection of ketamine/xylazine (150 and 10 mg/kg, respectively). Then, the sternum was cut and removed to expose the heart. The right atrium was immediately opened to sacrifice the animal. A 21G cannula was carefully inserted and clamped into the left ventricle through which 15 ml ice-cold PBS was perfused first. Then 15 ml ice-cold fixative (4% PFA in PBS) was perfused. The animal was decapitated, the brain was removed from the skull and kept in 4% PFA in PBS at 4˚C overnight. Then the brain was successively immersed in 10 and 30% sucrose solutions at 4˚C for cryoprotection until sunk.

The part of the brainstem containing the CN was fully covered by Tissue-Tek (Sakura Finetek, AJ Alphen aan den Rijn, The Netherlands). Parasagittal and frontal sections of 30 μm were cut using a cryotome (CM3050S, Leica Biosystems, Wetzlar, Germany). The CN sections were collected on coated slides (Scientific Menzel-Gläser Superfrost Plus) and double stained with the following steps. First, the sections were washed with PBS and incubated at RT for 3h with a blocking solution containing: 4% normal horse serum (NHS), 0.4% Triton X-100, 1% BSA in PBS. Second, the sections were incubated for 24h at 4˚C with a primary antibody solution containing: 1% NHS, 0.3% Triton X-100, 1% BSA in PBS and the primary antibodies. Third, after washing with PBS, the sections were incubated for 3h at RT with the secondary antibody solution containing: 0.02% Triton X-100, 1% BSA in PBS and the secondary antibodies. Finally, the sections were rinsed in PBS and stained with DAPI (1μg/ml in PB). Once coverslipped (in

Fluoprep, bioMérieux, UK) and sealed, the sections were stored in dark at 4˚C until imaging with a laser-scanning confocal microscope (TCS SP2, Leica microsystems, Germany). The contrast and the brightness of all images were adjusted using ImageJ (RRID: SCR_003070).

The primary antibodies used were: goat anti-calretinin antibody (1/500, Merck Millipore, Germany), rabbit anti-M1 muscarinic receptor (443–458) antibody (1/200, #AMR-010, RRID: AB_2340994, Alomone labs, Israel) and rabbit anti-M2 muscarinic receptor antibody (1/200, #AMR-002, RRID: AB_2039995, Alomone labs, Israel). The secondary antibodies used were: donkey anti-goat alexa488 (1/500, Life Technologies, USA) and donkey anti-rabbit alexa546 (1/500, Life Technologies, USA). The M1 staining was performed on a total of 7 slices and M2 staining on 13 slices.

### Statistics

We performed one- and two-way analysis of variance (ANOVA) with repeated measures using custom MATLAB (RRID: SCR_001622) code. For this we constructed repeated measure models from our data using the function *fitrm.m* and performed two-way ANOVA with repeated measures for the factors time/pulse number, pharmacological condition and the interaction of these factors. In the result section we report the F-statistics as F(a,b), with a and b being the degrees of freedom of factor and error, respectively. We never assumed sphericity and thus always use the conservative Greenhouse-Geiser correction of probability values. One-way ANOVA with repeated measures was performed accordingly for only one factor (pharmacological treatment). In case of significant outcomes of the repeated measure ANOVA, pairwise comparisons were performed with the MATLAB function *multcompare.m* for each factor. Here, probabilities were Tukey-Cramer corrected. Significance was determined for an α value of $<0.05$. In several occasions we also performed regular analysis of variance (ANOVA) and pairwise t-tests. Unless noted otherwise all data are expressed as mean ± standard error of the mean (SEM).

## Results

### Inhibition of M-current by oxotremorine-M in SBCs

In voltage clamp, both activation- and deactivation-, (or relaxation) protocols (upper parts in Fig 1A and 1B, respectively) permit measurement of $I_M$. For the former, the cell was held at -60 mV, and stepped to more depolarized values. In presence of Kv1.1/2 and HCN channel blockers, the slowly developing outward current (Fig 1A, lower part) reflected the slow opening of M-channels. In Fig 1A, $I_M$ was estimated at 690 pA at +10 mV step. For the deactivation protocol, the voltage was first clamped to -20 mV, which activated a substantial amount of M-channels and keep a constant outward $I_M$ (Fig 1B). Applying hyperpolarizing steps from this holding potential induced an instantaneous inward current followed by a slow inward current relaxation corresponding to the gradual closure of M-channels (Fig 1B and 1C). Please note that this protocol was thus designed to demonstrate non-inactivating, slow voltage activated conductances: other outward currents like A-type potassium currents would be inactivated by the high holding potential. Rapidly gated, non-inactivating outward currents, like Kv3-mediated potassium currents, are mostly deactivated during the "instantaneous" phase and thus do not pollute the slow relaxation current. A true identification of the non-inactivating, slow voltage activated outward current as M-current however must involve pharmacological experiments (see below). At -50 mV, the inward relaxation (difference between $I_{inst}$ and $I_{ss}$) was estimated at 96 pA. Stepping back to -20 mV resulted in a smaller instantaneous current ($I'_{inst}$) coincident with a reduction of the conductance caused by the closure of M-channels (Fig 1C). The slow opening of M-channels could be visualized by the slow outward tail $I_M$ current equal

to 193 pA at -50 mV (Fig 1C). For the most hyperpolarizing steps, a rapidly activating and rapidly inactivating outward current could be observed matching the kinetics of the potassium A-current (indicated by the arrow in Fig 1C).

In the activation protocol, measurements of $I_M$ are potentially contaminated by slow activation of other ion channels, unless a combination of blockers are added to the ACSF. The inward relaxation observed in the deactivation protocol however is considered to be mainly mixed with the activity of HCN channels. For this reason, the following quantification of $I_M$, before and after application of pharmacological agents, were only made with the deactivation protocol (Fig 2A) in the presence of ZD7288, a HCN channel blocker, in 16 cells (from gerbils aged from P14 to P25,cell capacitance = 23 ± 2 pF, IR = 156 ± 18 MΩ, Rs = 16.3 ± 1.5 MΩ).

The inward relaxation current was measured before and after wash-in of XE991, a blocker of Kv7 channels known to substantially contribute to $I_M$. The results for one example cell are presented in Fig 2. The maximum values of $I_M$ for this example cell in control conditions were measured at -50 and -40 mV steps for this cell, 108 and 111 pA, respectively (Fig 2C and 2D). The relaxation current was almost non-existent at -70 mV (Fig 2B and 2D) and reversed between -80 and -70 mV (not shown). The application of XE991 strongly reduced $I_M$ at all voltages (Fig 2B, middle part and 2D). Next, we applied oxoM which also induced a strong reduction (Fig 2E and 2F) from 74 to 19 pA at -50 mV. However, a substantial part of $I_M$ (20–30%) was still present at -50 and -40 mV after perfusion of both XE991 and oxoM. In this example, oxoM did not induce further substantial reduction in presence of XE991 (Fig 2B) since $I_M$ was almost entirely inhibited by XE991. Nonetheless, at -50 and -40 mV, a small additional reduction was discernable (Fig 2B, insets in middle and right parts and 2D).

In Fig 3A we show group data for the same experiments as described in Fig 2. On average, both step-voltage (F(4,16) = 6.35, p<0.05; n = 5) and XE991 treatment (F(1,4) = 14.69, p<0.05; n = 5) had a significant effect on $I_M$ amplitudes when tested with two-way ANOVA with repeated measures. There was also a significant interaction between the two factors (F(4,16) = 7.39, p<0.05; n = 5). Post-hoc pair-wise testing revealed significant differences between the control and the XE991 condition for the step voltages -40mV (p<0.05), -50mV (p<0.01) and -70mV (p<0.05). At -30mV (p = 0.09) and -60mV (p = 0.06) the differences were not statistically significant. The average of $I_M$ amplitude at -50 mV (n = 5) was reduced from 57 ± 15 pA to 5 ± 3 pA after washing in XE991. Noticeably, $I_M$ was almost completely abolished at more hyperpolarized voltage steps (-70 and -60 mV) whereas a persistent inward relaxation was recorded at -40 and -30 mV (17 ± 1 pA and 19 ± 4 pA, respectively, Fig 3A).

In Fig 3B we show group data for the muscarinic modulation of $I_M$ (cf. Fig 2F). On average, both step-voltage (F(4,28) = 19.6, p<0.001; n = 8) and oxoM treatment (F(1,7) = 32.8, p<0.001; n = 8) had a highly significant and very strong effect on $I_M$ amplitudes (two-way ANOVA with repeated measures). There was, again, a significant interaction between the two factors (F(4,28) = 8.42, p<0.001; n = 8). Post-hoc pair-wise testing revealed significant differences between the control and the oxoM condition for all step voltages (-30mV: p<0.01; -40mV: p<0.01; -50mV: p<0.001; -60mV: p<0.01) except -70mV (p = 0.16).

The small XE991-resistant current was not seen when oxoM was applied (Fig 3B). In presence of XE991 (n = 5), further change of inward relaxation caused by oxoM was minimal or none (Fig 3C). This is reflected in the statistical analysis comparing the XE991 condition with the XE991+oxoM condition: neither voltage (F(4,16) = 0.86, p = 0.41) nor pharmacological condition (F(1,4) = 7.41, p = 0.053) was a significant factor, nor was any interaction between the factors left (F(4,16) = 2.36, p = 0.17).

In Fig 3D, for each agent tested, the percentage of $I_M$ inhibition was calculated (% inhibition = (1-($I_{test}/I_{ctrl}$)) x 100) at -50 mV to further illustrate our findings. In control condition, XE991 and oxoM inhibited $I_M$ by 94 ± 6% and 82 ± 7%, respectively (ANOVA p<0.001, df = 2, F = 13.76;

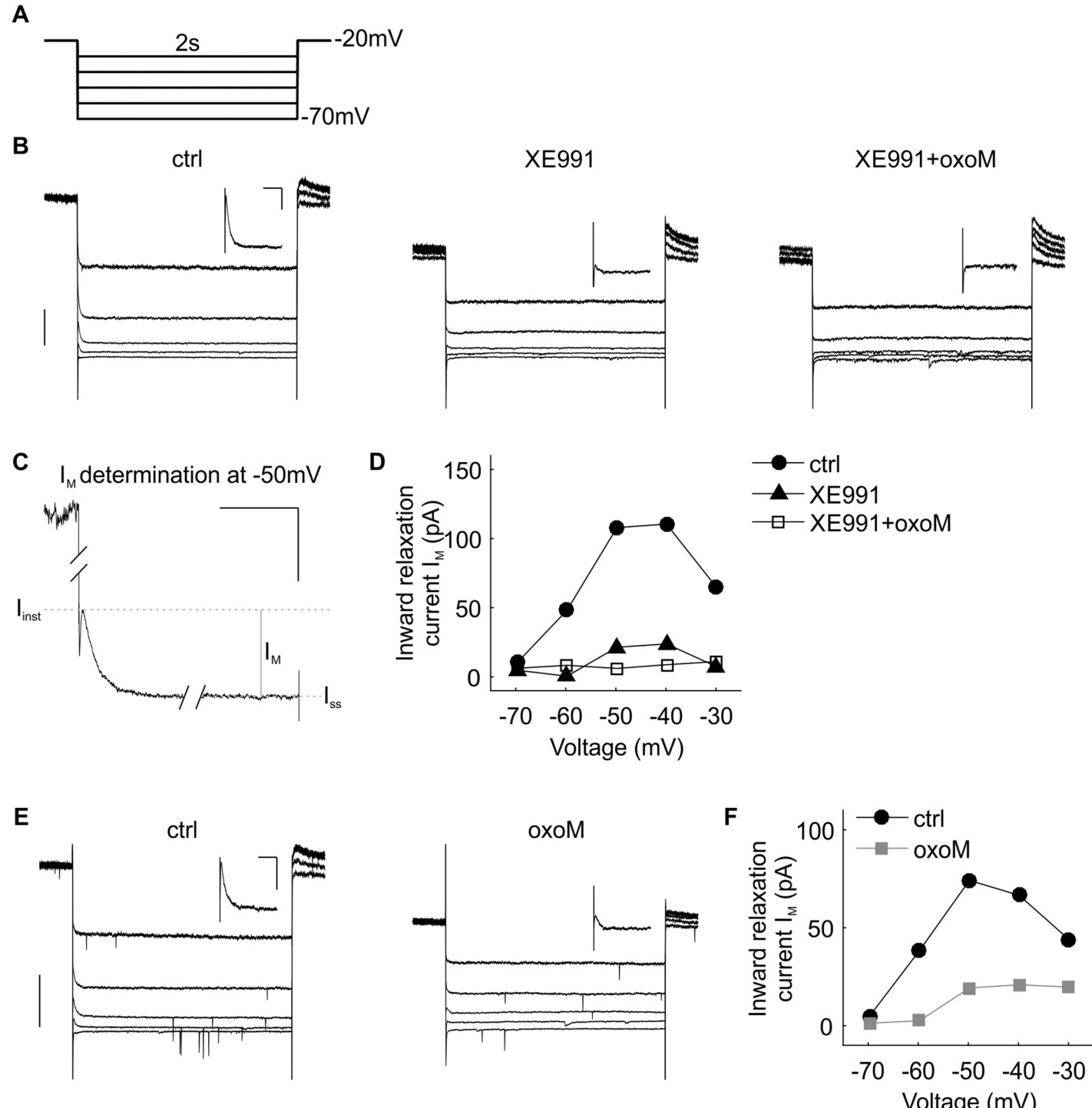

**Fig 2. $I_M$ suppression by oxoM in an example cell.** (A) Deactivation protocol with voltage steps (2 s) from -70 mV to -30 mV. (B) Example traces of current recordings in control (ctrl) condition (left part), after XE991 wash-in (middle part) and application of oxoM in presence of XE991 (right part); scale: 200 pA. The insets on the top right are magnifications of the -50 mV step; scale: 100 ms/50 pA. (C) Higher magnification of the -50 mV step in B (left part) showing how the inward relaxation, $I_M$, was measured, i.e. by the difference between instantaneous $I_{inst}$ and steady state $I_{ss}$ currents; scale 100 ms/100 pA. A clear reduction of the inward relaxation was observed between the ctrl and XE991 conditions at -50 mV, see insets in B left and middle parts, while no further substantial change was noticed when oxoM is applied (B, right part). (D) Values of $I_M$ at each voltage for the cell shown in B. (E) Example traces of current recordings in control condition (left part) and after application of oxoM (right part); scale: 200 pA. The insets are the magnification of the -50 mV step; scale: 100 ms/50 pA. The inward relaxation $I_M$ was clearly reduced by oxoM. (F) Values of $I_M$ at each voltage for the cell shown in E.

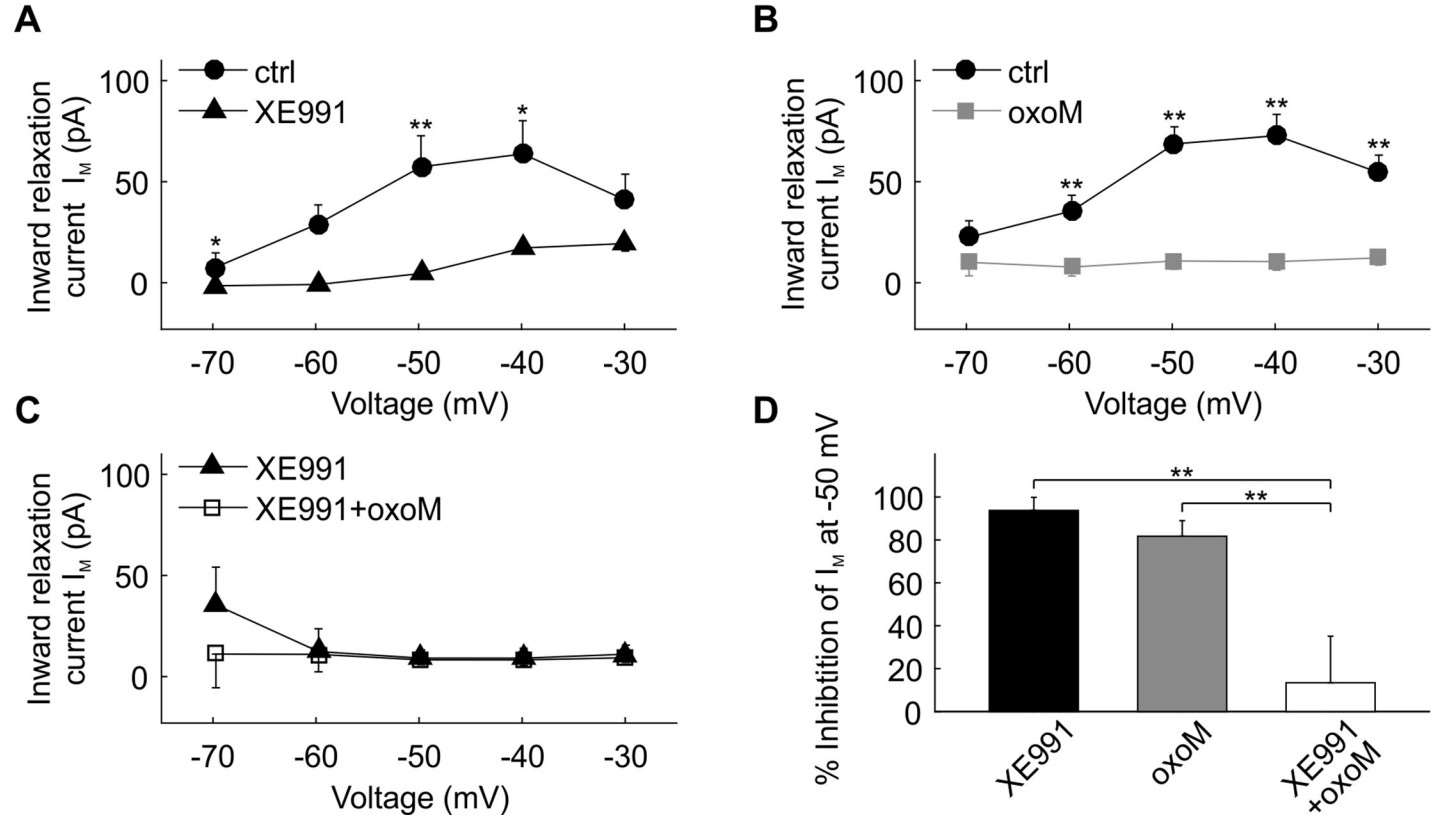

**Fig 3. Quantification of $I_M$ suppression by oxoM in SBCs.** (A) Voltage-current relationship of the mean of the inward relaxation current $I_M$ (n = 5) in control and after application of XE991. (B) Voltage-current relationship of the mean of the inward relaxation current $I_M$ (n = 8) in control and after application of oxoM. (C) Voltage-current relationship of the mean of the inward relaxation current $I_M$ (n = 5) in presence of XE991 and after application of oxoM. (D) Percentage of $I_M$ at -50 mV inhibited by the application of XE991 (black), oxoM (grey) and oxoM in presence of XE991 (white). Application of XE991 or oxoM alone strongly reduced the inward relaxation current $I_M$ at -50 mV. Application of oxoM only inhibited a small amount of the residual $I_M$ when the cell was pre-treated with XE991. * p<0.05; ** p<0.01.

post-hoc test p>0.9999 XE991 (black) vs. oxoM (grey)). On the other hand, oxoM application in presence of XE991 had a weak effect on residual $I_M$ (13 ± 22%) compared to the other conditions (post hoc p<0.01, oxoM vs. XE991+oxoM; p<0.001, XE991 vs. XE991+oxoM), indicating a broad overlap of affected conductance between XE991 and oxoM.

The strong reduction of the inward relaxation by XE991 application showed the presence of $I_M$ most likely caused by Kv7 (KCNQ-type) channels in SBCs. This current was strongly modulated by the application of a muscarinic agonist suggesting that $I_M$ plays a role in the cholinergic influence on SBCs.

### Inhibition of $I_h$ by oxotremorine-M in SBCs

$I_h$ current was recorded (Fig 4A) in presence of XE991 to avoid contamination by $I_M$. $I_h$ is mediated by the family of hyperpolarization-activated cyclic nucleotide-gated channels (HCN) and can be blocked by ZD7288 (Fig 4B, left and middle parts). The amplitude of $I_h$ was measured in 14 cells (from gerbils aged from P19 to P25, cell capacitance = 18 ± 1pF; IR = 102 ± 8 MΩ, Rs = 19.9 ± 1.4 MΩ). The slow inward current, mostly present at voltages more negative than -60 mV (Fig 4B), reflected the slow opening of HCN channels. $I_h$ was almost completely abolished (from 402 to 46 pA at -110 mV) after applying ZD7288 for at least 10 minutes (Fig 4B, middle part). No further noteworthy change was observed after application of oxoM (Fig

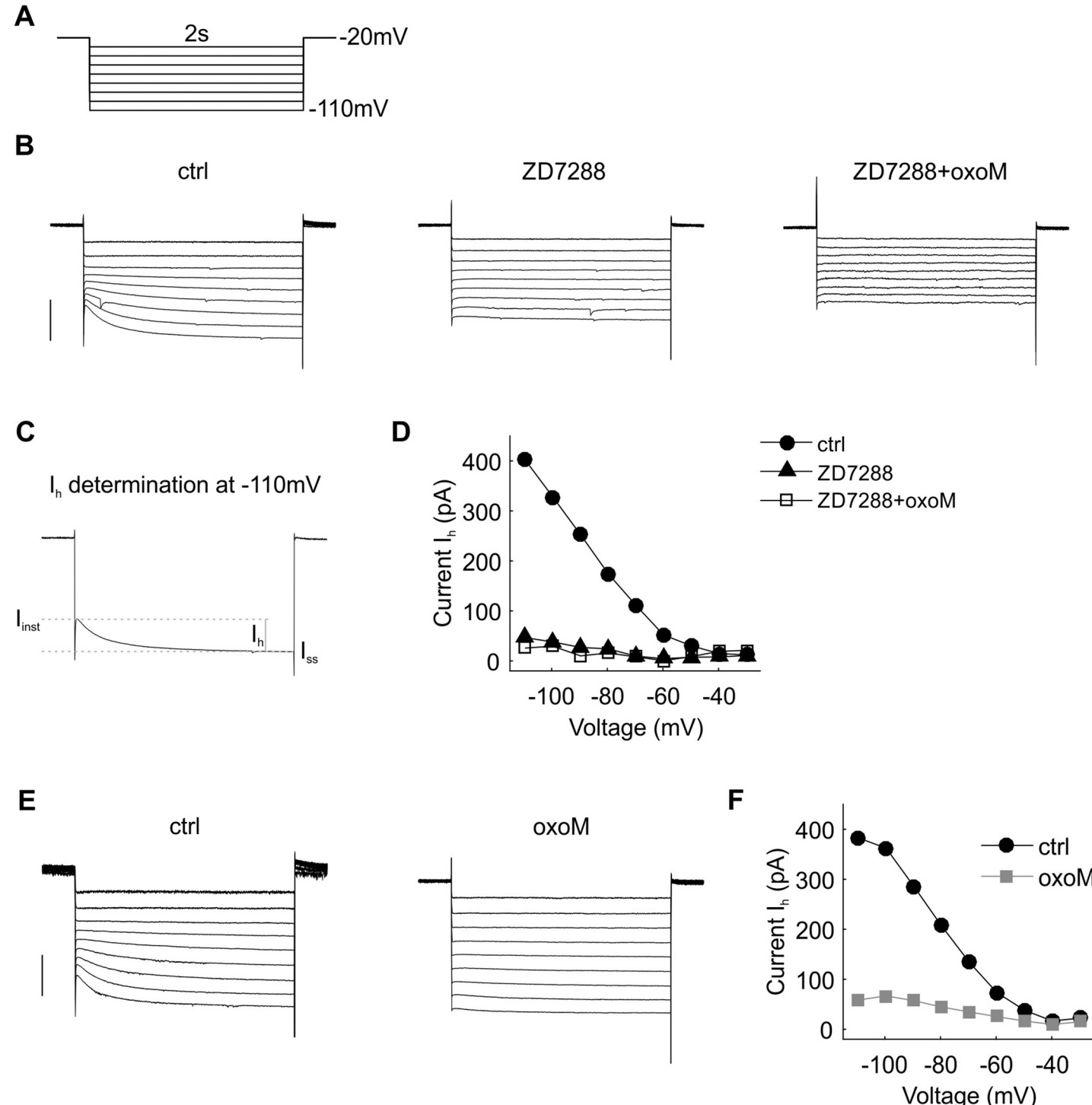

**Fig 4. $I_h$ suppression by oxoM in an example cell.** (A) Deactivation protocol with voltage steps (2 s) from -110 mV to -30 mV. (B) Example traces of current recordings in control (ctrl) condition (left part), after ZD7288 wash-in (middle part) and application of oxoM in presence of ZD7288 (right part); scale: 500 pA. (C) Step at -110 mV from B (left part) showing how the inward current $I_h$ was measured. $I_h$ was clearly visible at the more hyperpolarized voltage steps in ctrl but was almost completely gone in presence of ZD7288. (D) Values of $I_h$ at each voltage for the cell shown in B. (E) Example traces of current recordings in control condition (upper part) and after application of oxoM; scale: 500 pA. (F) Values of $I_h$ at each voltage for the cell shown in E. $I_h$ was also considerably reduced by the application of oxoM alone.

4B, right part). In contrast to this, in absence of ZD7288 oxoM prominently decreased $I_h$ amplitude from 382 to 59 pA at -110 mV (Fig 4E). Fig 4E and 4F summarize the effect of ZD7288 and oxoM on $I_h$ amplitude for each voltage step for this example cell. The profile of the two IV curves was similar in response to the two agents tested. At -30 and -40 mV in control condition, $I_h$ was negligible (17 and 22 pA in Fig 4D and 15 and 12 pA in Fig 4F, respectively). In the presence of either ZD7288 or oxoM, $I_h$ was strongly reduced at each voltage but the most robust decrease was observed at the highest hyperpolarized value tested ($I_h$ reduced by 8 times with ZD7288 and 7 times with oxoM, see Fig 4D and 4F). Fig 4D shows that there was no further effect of oxoM on membrane currents upon hyperpolarizing voltage steps when ZD7288 was already present in the ACSF.

Group results for the same experiments as shown in Fig 4 are presented in Fig 5. The average amplitudes of $I_h$ (Fig 5A) were significantly influenced by voltage ($F_{(8,32)} = 50.23$, $p < 0.001$; $n = 5$) and ZD7288 ($F_{(1,4)} = 80.9$, $p < 0.001$; $n = 5$). Post-hoc pair-wise comparisons revealed, that the reduction of $I_h$ by ZD7288 was statistically significant and strong for step voltages of -70 mV (reduction of -129 pA ± 14.7 pA, $p < 0.001$) and all more hyperpolarized values (-80 mV: -178 ± 19.5 pA, $p < 0.001$; -90 mV: -261 ± 31 pA, $p < 0.01$; -100 mV: -319 ± 29.3 pA, $p < 0.001$; -110 mV: -375 ± 46.5 pA, $p < 0.01$).

OxoM application (Fig 5B) on average also strongly reduced H-currents in $n = 5$ spherical bushy cells. Again, both voltage ($F_{(8,32)} = 34.9$, $p < 0.01$; $n = 5$) and oxoM ($F_{(1,4)} = 26$,

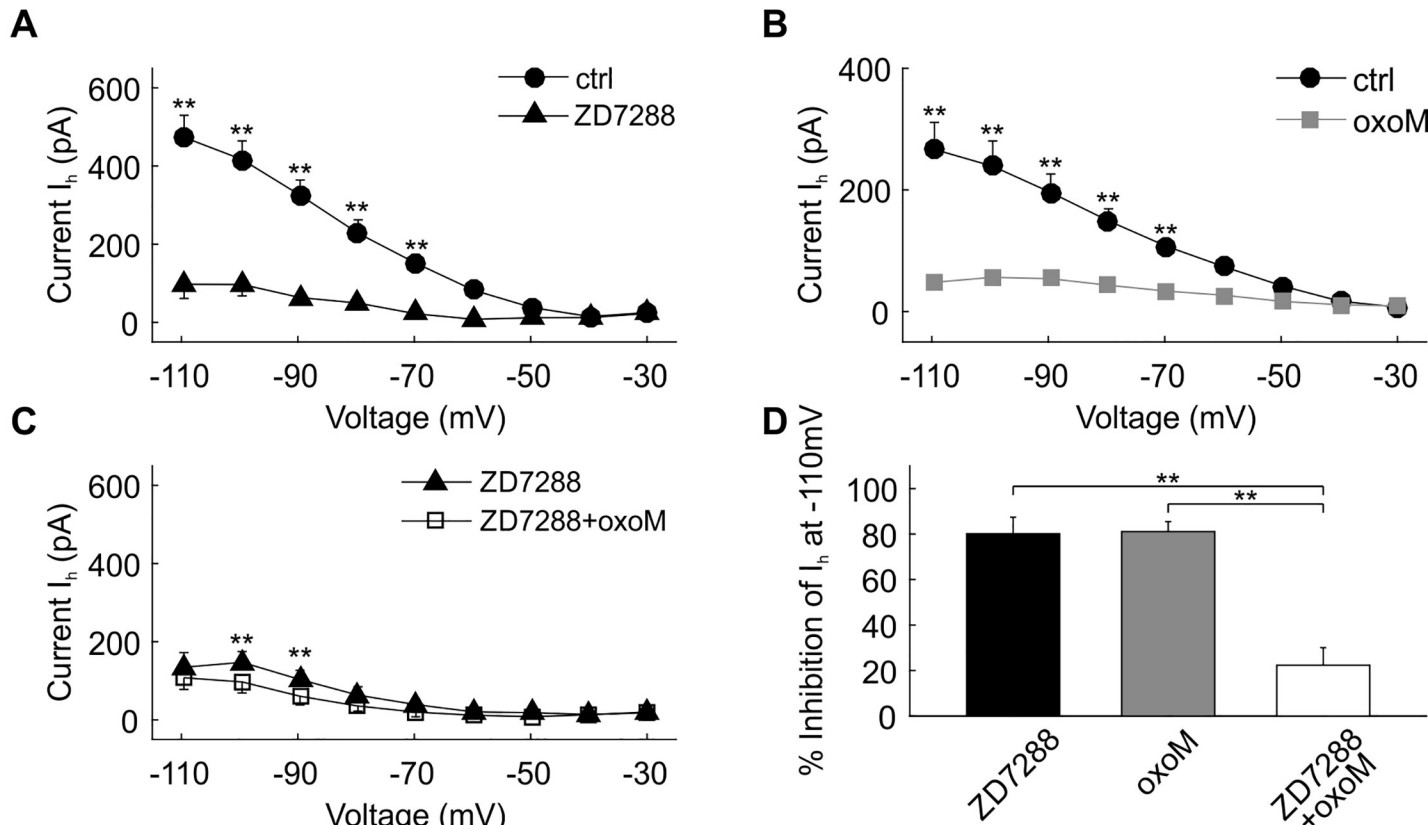

**Fig 5. Quantification of $I_h$ suppression by oxoM in SBCs.** (A) Voltage-current relationship of the mean of the current $I_h$ ($n = 5$) in control and after application of ZD7288. (B) Voltage-current relationship of the mean of the current $I_h$ ($n = 5$) in control and after application of oxoM. (C) Voltage-current relationship of the mean of the current $I_h$ ($n = 6$) in presence of ZD7288 and after application of oxoM. (D) Percentage inhibition of $I_h$ at -110 mV induced by the application of ZD7288 (black) or oxoM (grey) in control condition or by the application of oxoM in presence of ZD7288 (white). Application of ZD7288 or oxoM alone significantly reduced $I_h$ at -110 mV. Application of oxoM did not reduce $I_h$ when the cell was pre-treated with ZD7288. * $p < 0.05$; ** $p < 0.01$.

p<0.01; n = 5) significantly influenced $I_h$. Again, there was a significant interaction between the factors (F(8,32) = 43.6, p<0.001). Similar to the results for the ZD7288 experiment, oxoM significantly reduced $I_h$ for step voltages of -70 mV (reduction of -73 ± 8.7 pA, p<0.01) and lower (-80 mV: -105 ± 15 pA, p<0.01; -90 mV: -141 ± 24 pA, p<0.01; -100 mV: -182 ± 32 pA, p<0.01; -110 mV: -218 ± 36 pA, p<0.01).

The presence of ZD7288 before oxoM application prevented large effects of oxoM on $I_h$ (Fig 5C). When we tested the ZD7288 vs. the ZD7288+oxoM conditions in two-way ANOVA with repeated measures, voltage was still a weak but significant factor for $I_h$ (F(8,40) = 13.6, p<0.01; n = 6) but not pharmacological condition (F(1,5) = 1.18, p = 0.32; n = 6). A significant interaction between the factors (F(8,40) = 9.6, p<0.01; n = 6) was apparent. This finding was supported by the post-hoc pair-wise testing: For voltages of -90 mV (reduction of -19 ± 8.1 pA, p<0.01) and -100 mV (reduction of -50 ± 16 pA, p<0.05) oxoM further reduced the average currents, suggesting incomplete block of $I_h$ by ZD7288 or an oxoM effect on leak channels that are ZD7288 insensitive. We will further discuss this finding in a later section of this paper.

At -110 mV, ZD7288 and oxoM similarly inhibited $I_h$ by 80 ± 7% and 81 ± 4%, respectively (Fig 5D, ANOVA p<0.0001, df = 2, F = 31.3; post-hoc test p>0.9999 ZD7288 vs. oxoM) while the effect of oxoM in presence of ZD7288 was considerably reduced, 22 ± 8% (oxoM (grey) vs. ZD7288+oxoM, p<0.0001; ZD7288 vs. ZD7288+oxoM, p<0.0001).

Overall these data showed that $I_h$ was strongly decreased by the application of a muscarinic agonist in SBCs. This suggests that both $I_M$ and $I_h$ could potentially play a role as effectors in the cholinergic modulation of spherical bushy cells.

## Expression of M1 and M2 muscarinic receptors in AVCN

The expression of M3 receptor, known to modulate the M-current, was already shown in the neuropil of gerbil SBCs [5]. However, several studies have shown that M1 also played a role in $I_M$ [26–30]. In addition, $I_h$ can be modulated by the intracellular signaling pathway triggered by the $G_{i/o}\alpha$ protein, which is activated by M2 and M4 muscarinic receptors. For this reason, the presence of other muscarinic receptors was evaluated in parasagittal slices of the CN. Here we present the results obtained for M1 and M2. The calretinin-positive auditory nerve terminals helped to differentiate the SBCs (filled arrows) in the AVCN from other cell types (Fig 6A). A strong M1 signal was uniformly present in the entire AVCN and localized on cell bodies (Fig 6B). All AVCN neuronal cell bodies seemed to present the M1 staining suggesting that SBCs are not the only cell type expressing M1 in the AVCN (Fig 6C and 6D, empty arrows).

Fig 6E and 6F show calretinin and M2 staining, respectively. Contrary to M1, M2 appeared weaker, more intense near the lateral edges of AVCN and mainly localized in the neuropil of AVCN cells. Few intense M2 signals could be noticed around cell bodies (Fig 6F, 6G and 6H, filled arrowheads) but no co-localization with calretinin-positive endbulbs was observed. In addition, sparse M2-positive puncta were observed on SBC cell bodies (Fig 6H, empty arrowheads) implying that M2 acts on the postsynaptic site. This suggests that M2 in the gerbil AVCN might not function as heteroceptor on the presynaptic site, as was previously reported [31] for AN terminals.

These results confirmed the presence of both M1 and M2 muscarinic receptors in AVCN. Staining against M1 was stronger and mainly found on the postsynaptic site of both SBCs and non-SBCs while M2 was predominantly localized in the neuropil surrounding SBCs. In addition to the M3 muscarinic receptor expression we have shown before [5], SBCs express a variety of muscarinic receptors of partially overlapping function. The non-uniform distribution of M2 signals throughout the AVCN suggests the possibility of a diversity of muscarinic responses in SBC.

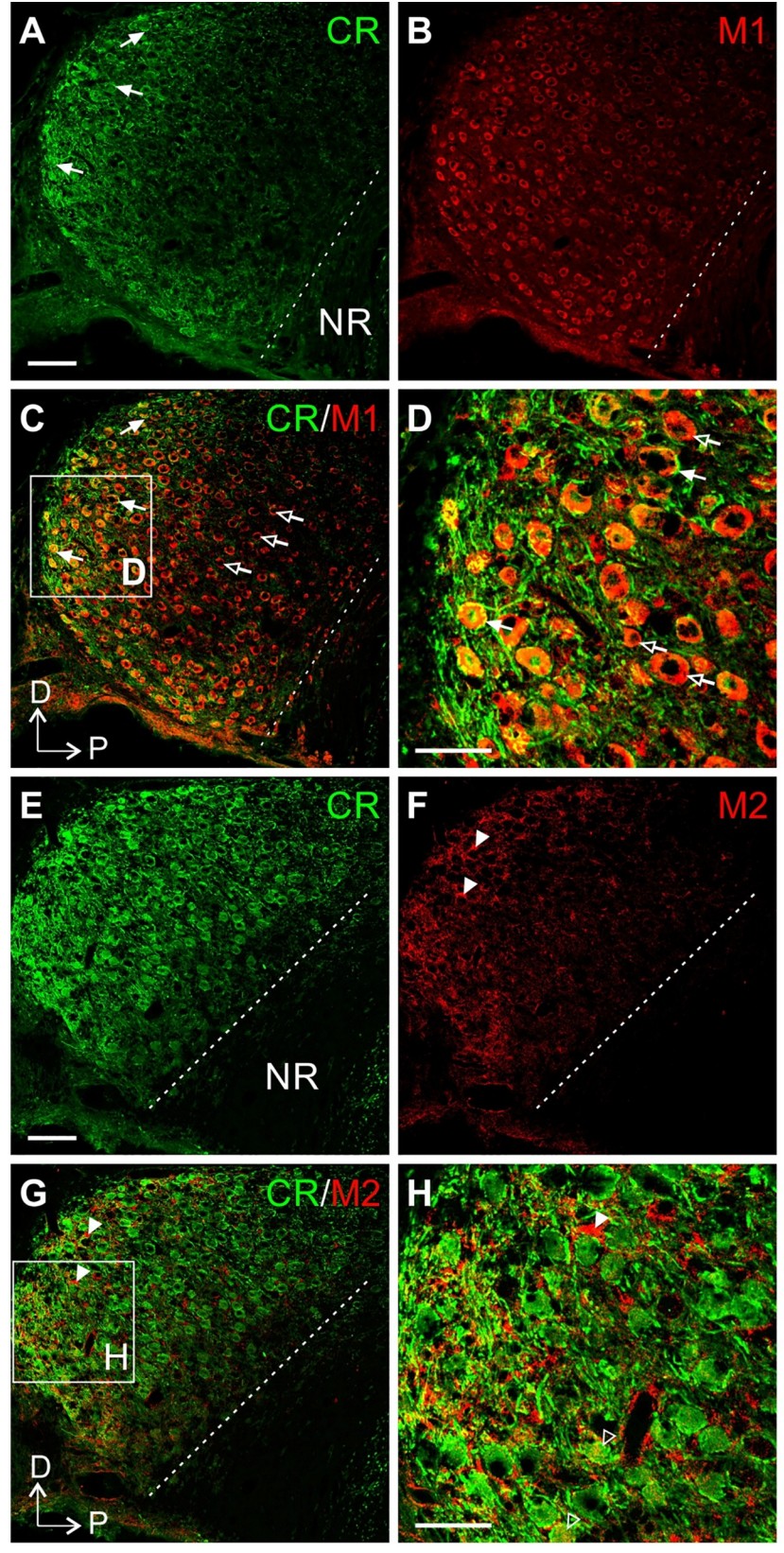

**Fig 6. Expression of M1 and M2 muscarinic receptors in gerbil AVCN.** (A) Calretinin (CR) staining in a parasagittal slice of gerbil cochlear nucleus. The filled arrows point to strong calretinin staining corresponding to big auditory nerve fiber terminals, the endbulb of Held. Scale: 100 μm. (B) M1 muscarinic staining was found on all AVCN cell bodies. (C) Merged image of A and B. Filled arrows show M1-positive SBCs. Empty arrows show M1-positive cells which are not SBCs. (D) Digital magnification of the white square in C. Scale: 50 μm. (E) Calretinin staining in a parasagittal slice of gerbil cochlear nucleus in which endbulb of Held could be identified. Scale: 100 μm. (F) M2 muscarinic staining was mostly present in the anterior part of AVCN, around SBCs. Filled arrowheads indicate sparse but strong M2 staining close to cell bodies. (G) Merged image of E and F. (H) Digital magnification of the white square in G. Empty arrowheads point to M2-positive puncta on SBC cell bodies. Scale: 50 μm (magnification: 63x). D = dorsal; P = posterior; NR = nerve root.

## No effect of carbachol on synaptic transmission at the endbulb of Held

Since immunohistochemical results from our previous work pointed out evidence of presynaptic nicotinic receptors [5,10] and we showed the expression of M2 in AVCN (which can act as presynaptic heteroceptor), the effect of a cholinergic agonist on the synaptic transmission between auditory nerve and SBC was evaluated. For that, synaptic transmission was investigated by recording evoked EPSCs (n = 6, from gerbils aged from P15 to P17, cell capacitance = 33 ± 4 pF, IR = 234 ± 16 MΩ, Rs = 10.9 ± 1.2 MΩ) and spontaneous miniature EPSCs (n = 8, from gerbils aged from P16 to P20, cell capacitance = 30 ± 3 pF, IR = 276 ± 50 MΩ, Rs = 12.3 ± 1.1 MΩ) before and after 10 minutes of carbachol application.

A frequency-dependent depression of evoked EPSC amplitudes (Fig 7B1-3, n = 6) was evident in our recordings. Even though in the 200 Hz conditions many missing values (which represent insufficient EPSC at that given pulse number) made statistical analysis difficult, the pulse number generally had a highly significant influence on EPSC amplitude (two-way ANOVA with repeated measures, 50 Hz: $F_{(29,145)} = 45.2$, $p<0.001$; 100 Hz: $F_{(29,145)} = 52.7$, $p<0.001$; 200 Hz: $F_{(29,29)} = 37.9$, $p = 0.1$). However, depression of EPSC amplitudes was not statistically different between the control and the carbachol groups (50 Hz: $F_{(1,5)} = 0.89$, $p = 0.38$; 100 Hz: $F_{(1,5)} = 1.56$, $P = 0.26$; 200 Hz: $F_{(1,1)} = 1.5$, $p = 0.43$) and very weak or no significant interaction between pulse-number and pharmacological treatment was observed (50 Hz: $F_{(29,145)} = 3.25$, $p = 0.04$; 10 0Hz: $F_{(29,145)} = 2.51$, $p = 0.15$; 200 Hz: $F_{(29,29)} = 1.34$, $p = 0.45$). The 10%-90% rise time of $eEPSC_{1-15}$ and the decay time constant of $eEPSC_{1-15}$ were determined for all frequencies and in both conditions (Fig 7B4-6 for rise time, 7B7-9 for decay time constant). For the 10%-90% rise time we found a robust effect of pulse number (two-way ANOVA with repeated measures, 50 Hz: $F_{(13,65)} = 15.9$, $p<0.001$; 100 Hz: $F_{(13,65)} = 11.7$, $p<0.01$; 200 Hz: $F_{(13,39)} = 12.1$, $p<0.01$), however we did not find a statistically significant influence of pharmacological condition (50 Hz: $F_{(1,5)} = 0.06$, $p = 0.8$; 100 Hz: $F_{(1,5)} = 0.002$, $p = 0.96$; 200 Hz: $F_{(1,3)} = 0.68$, $p = 0.46$) or any interactions (50 Hz: $F_{(13,65)} = 0.38$, $p = 0.79$; 100 Hz: $F_{(13,65)} = 0.41$, $P = 0.65$; 200 Hz: $F_{(13,39)} = 2.1$, $p = 0.22$). For the decay time constant of $EPSC_{1-15}$, neither pulse number, pharmacological condition nor the interaction of the factors had a statistically significant effect when tested with a two-way ANOVA with repeated measures.

In addition to EPSC dynamics we estimated the amount of asynchronous release of the synapse during steady-state stimulation at three different stimulation frequencies (Fig 7C). While a highly significant increase of estimated asynchronous release with frequency was evident (two-way ANOVA with repeated measures, $F_{(2,10)} = 218$, $p<0.001$), pharmacological condition had no influence ($F_{(1,5)} = 0.54$, $p = 0.49$), neither did we find any interaction of factors ($F_{(2,10)} = 0.25$, $p = 0.7$).

Finally we wanted to roughly estimate the release probability of the synapse at the different conditions. For this we estimated the readily-releasable pool from our pulse-train data with a back-extrapolation method and expressed the release probability as the fraction of the pool (in

nA) expended by EPSC$_1$ (Fig 7D). Neither stimulation frequency, nor pharmacological condition or the interaction of the factors had a statistically significant effect on the release probability when tested with a two-way ANOVA with repeated measures.

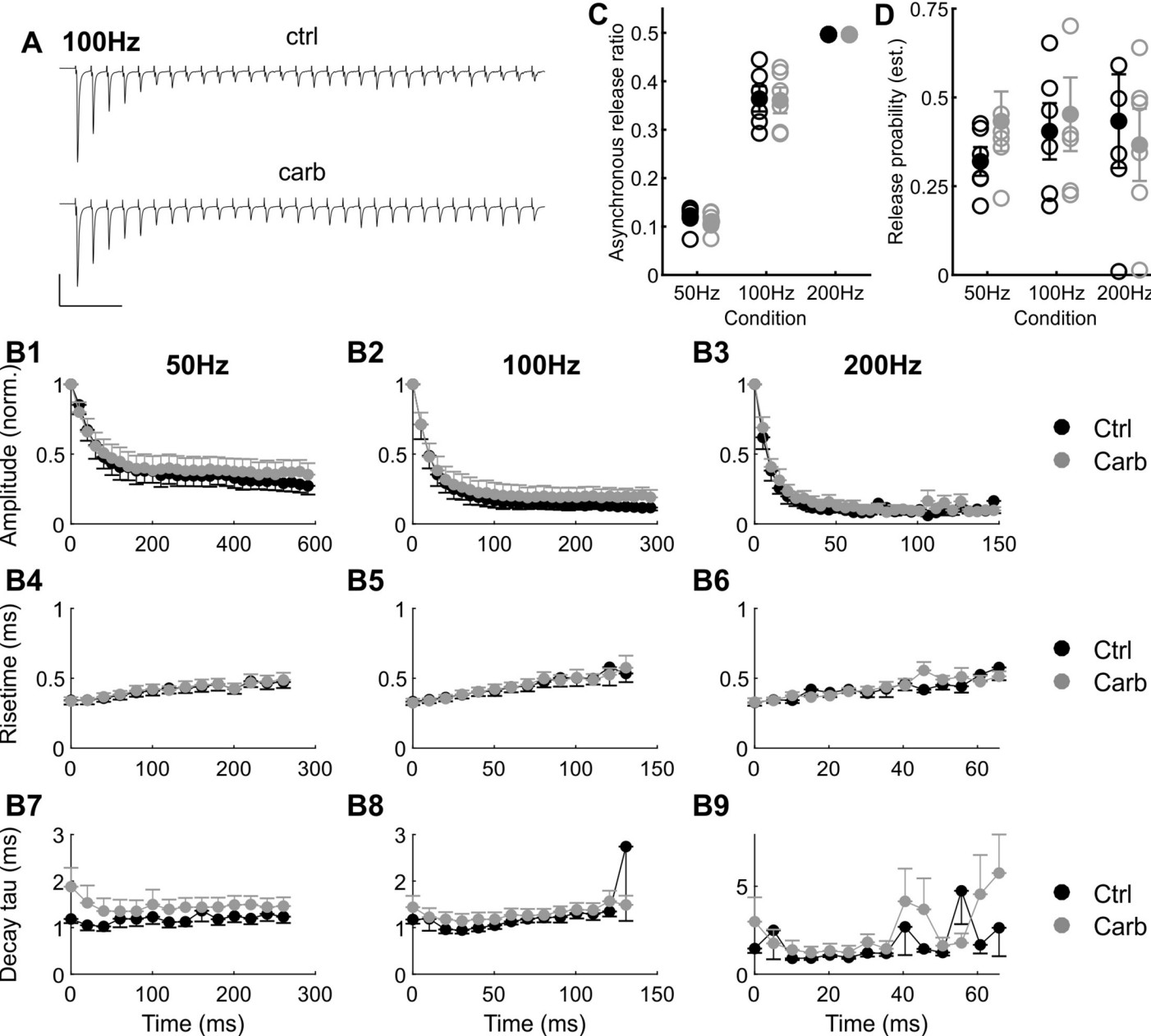

**Fig 7. No effect of carbachol on evoked EPSCs and short term depression.** (A) Example traces of recorded eEPSCs at 100Hz during the 30 stimulus train in control (ctrl) condition (upper part) and in presence of carbachol (carb, lower part); scale: 50 ms/3 nA. In both cases, a strong short-term depression was observed. (B1-3) Normalized average amplitude of eEPSCs for 50 Hz (B1), 100 Hz (B2) and 200 Hz (B3) compared between control (black) and carbachol (grey) conditions. No significant effects of carbachol were found. n = 6. (B4-6) Rise time of eEPSCs for 50 Hz (B1), 100 Hz (B2) and 200 Hz (B3) compared between control (black) and carbachol (grey) conditions. No significant effects of carbachol were found. n = 6. (B7-9) Decay time constant of eEPSCs for 50 Hz (B1), 100 Hz (B2) and 200 Hz (B3) compared between control (black) and carbachol (grey) conditions. No significant effects of carbachol were found. n = 6. (C) Estimated amount of asynchronous release (given as the amount of area under the curve of the second half of the stimulus duty cycle, divided by the area under the curve of the first half of the stimulus duty cycle) steeply increased with stimulus frequency but is not affected by carbachol. Please note: individual values (open circles) for the 200Hz condition masked by the marker for the group mean (closed circles). (D) The estimated release probability of the endbulb of Held (given as the amount of current released by the first EPSC divided by the estimated size of the readily-releasable pool) does not significantly depend on stimulus frequency or carbachol.

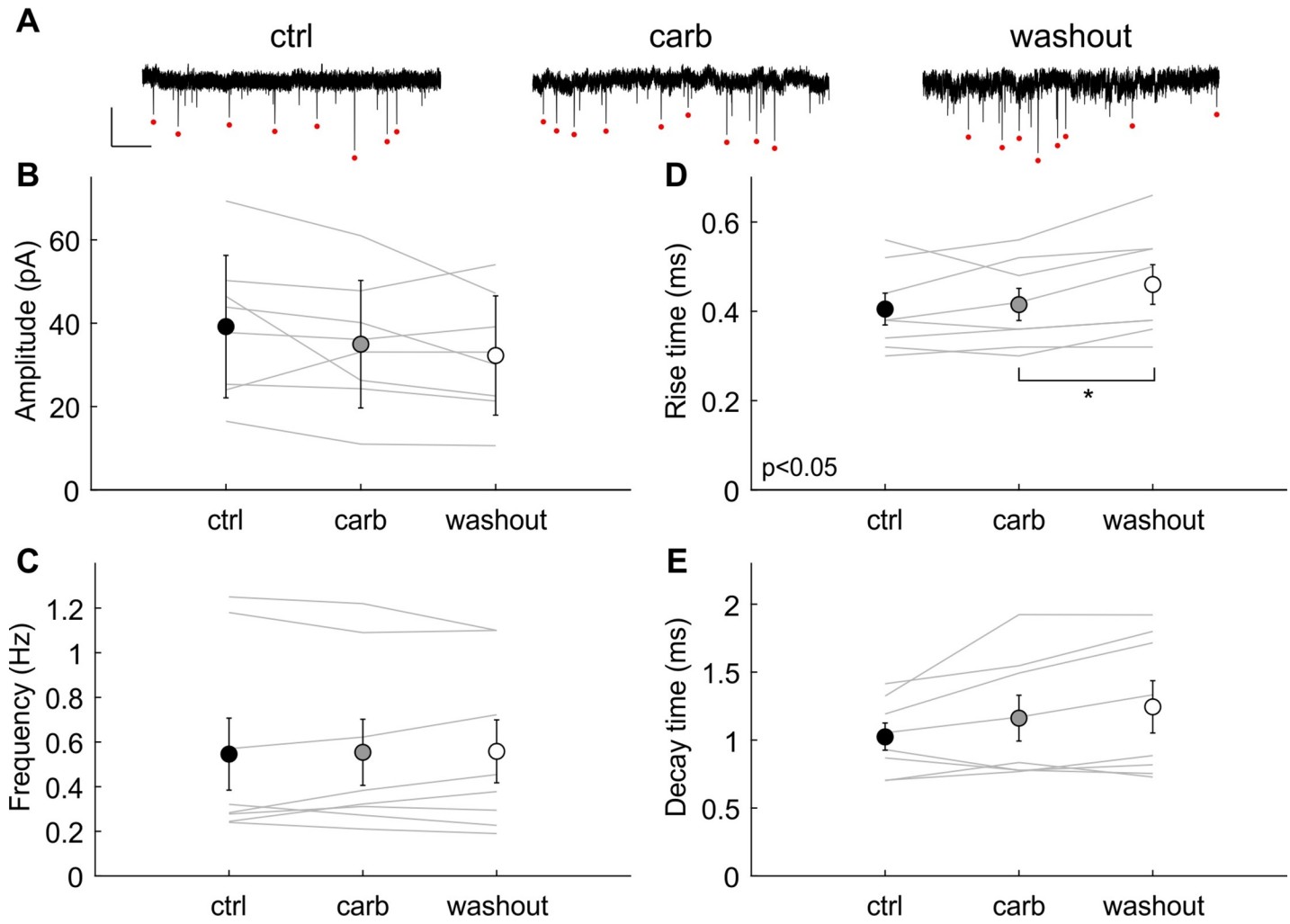

**Fig 8. No influence of carbachol on spontaneous mEPSCs.** (A) Example traces of recorded mEPSCs in control condition (ctrl, left part), in presence of carbachol (carb, middle part) and after washout carbachol (washout, right part); scale: 2 s/50 pA. The red dots represent the automatically detected mEPSCs. (B) Average amplitude, (C) average frequency, (D) average rise time and (E) average decay time of mEPSCs before, during and after washing out carbachol. No difference was observed between the three conditions for amplitude and rise- or decay kinetics. Although there was a significant difference between carbachol and wash-out for the mEPSC rise-time (asterisk in D) the effect was only very small.

We therefore concluded, that we could not find any significant influence of cholinergic signalling on endbulb of Held synchronous and asynchronous evoked synaptic transmission in the gerbil.

Next we wanted to analyze the effect of carbachol on spontaneous release of vesicles at the endbulb of Held and thus recorded spontaneous miniature synaptic currents (mEPSC). The mEPSCs were recorded during at least 5 minutes. Example traces are shown in Fig 8A. The average amplitude, the frequency and the kinetics of mEPSCs were measured before, during and after applying carbachol. The mean mEPSC amplitude was 39 ± 17 pA (n = 8) in control conditions. No significant change was observed in presence of carbachol (35 ± 15 pA, n = 8) and after wash-out (32 ± 14 pA; Fig 8B, n = 8; one-way ANOVA with repeated measures: $F_{(2,14)}$ = 2.29, p = 0.16). The average frequency was 0.55 ± 0.16 Hz (n = 8) in control condition and remained at, 0.55 ± 0.15 Hz (n = 8) in the presence of carbachol and 0.56 ± 0.14 Hz after wash-out (Fig 8C, $F_{(2,14)}$ = 0.07, p = 0.82). The mean 10%-90% rise time was 405 ± 88 μs

(control, n = 8), 415 ± 89 μs (carbachol, n = 8) and 460 ± 110 μs (wash-out, n = 8, Fig 8D). There was a significant difference between the wash-out and the carbachol condition (p<0.05 after one-way ANOVA with repeated measures, F(2,14) = 5.86, p<0.05, post-hoc t-tests are Bonferroni corrected). However, the mean effect size was very small and there was no difference between the control and the wash-out condition (p = 0.07). The mean time constants of mEPSC decay were 1.02 ± 0.25 ms (control, n = 8), 1.16 ± 0.41 ms (carbachol, n = 8) and 1.24 ± 0.47 ms (wash-out, n = 8, Fig 8E), differences were not significant (F(2,14) = 4.1, p = 0.06). Overall we conclude that there was no influence of carbachol on spontaneous vesicle release at the endbulb of Held of the gerbil.

Taken together, our measurements of evoked and spontaneous synaptic events at the endbulb of Held suggested that cholinergic transmission in the AVCN predominantly affected the excitability of the spherical bushy cell and did not significantly affect synaptic release or short term plasticity.

## Slow muscarinic effects on SBC membrane potential and excitability

The application of oxoM decreased $I_M$ and $I_h$ currents but the physiological meaning of this was unclear. In a set of current clamp experiments (Fig 9 & Fig 10), functional physiological parameters were monitored over several minutes every 30 s before, during and after bath application of carbachol (broad cholinergic agonist), oxoM or 4DAMP (M3 antagonist) in a total of 34 cells (from gerbils aged from P14 to P22; cell capacitance = 22 ± 2 pF; IR = 78 ± 3 MΩ, Rs = 15.8 ± 1.5). The average RMP in control condition was -61.95 ± 0.36 mV.

Fig 9A shows examples of voltage traces under injection of 10 current steps comprised between -0.3 to 0.3 nA (adjusted according the cell recording) before the application of carbachol (left), after 3 min (middle) and after 10 min (right) of carbachol perfusion. At suprathreshold current injection, a single action potential was observed at the beginning of the step in each condition. The initial resting potential for this cell was measured at –62 mV (dashed grey line). After 3 min in presence of carbachol, the resting potential was shifted to –59 mV and came back to the –62 mV even before washing out carbachol. The IV curve for the 3 recordings shown in Fig 9A was plotted in Fig 9B and used to estimate the RMP and the IR. The average change in RMP was plotted against time in Fig 9C (upper part, n = 5, shaded areas = SE). The thicker black line on the X axis represents the duration of carbachol application. Carbachol induced a depolarization reaching a maximum of +3.0 ± 1.1 mV (n = 5, Fig 9C, lower part) at the beginning of the perfusion and the RMP came back to the initial value before the end of carbachol application. Nevertheless, this change failed to be significant when comparing the maximum change (one-way ANOVA with repeated measures F(2,8) = 3.50, p = 0.06, n = 5, Fig 9C, lower part). This depolarization was accompanied by a significant change of –7.7 ± 4.7 MΩ (n = 5) in IR (F(2,8) = 4.55, p<0.05, n = 5; post-hoc test: ctrl vs. carb, p<0.05, Fig 9D). This transient decrease in IR, reflecting an increase in conductance, could be explained by the opening of nicotinic receptors. Noteworthy, after getting back to the initial value, the IR continued to increase until the end of the recording. This was associated with an instable and weak depolarization (Fig 9C, upper part) that could be mediated by muscarinic receptors. As a control, the same protocol was applied but the cholinergic agent was omitted (Fig 9E and 9F). No significant change was observed in both RMP (F(2,6) = 0.48, p = 0.57, n = 4) and IR (F(2,6) = 0.06, p = 0.85, n = 4).

In Fig 9G (upper part), the resting potential distinctly hyperpolarized upon oxoM perfusion reaching a max change of -2.4 ± 1.0 mV and went back to the initial value when oxoM was washed out. This was marginally significant in one-way ANOVA with repeated measures (F (2,28) = 4.9, p<0.05; n = 15, Fig 9G, lower part), however variability was high due to opposite

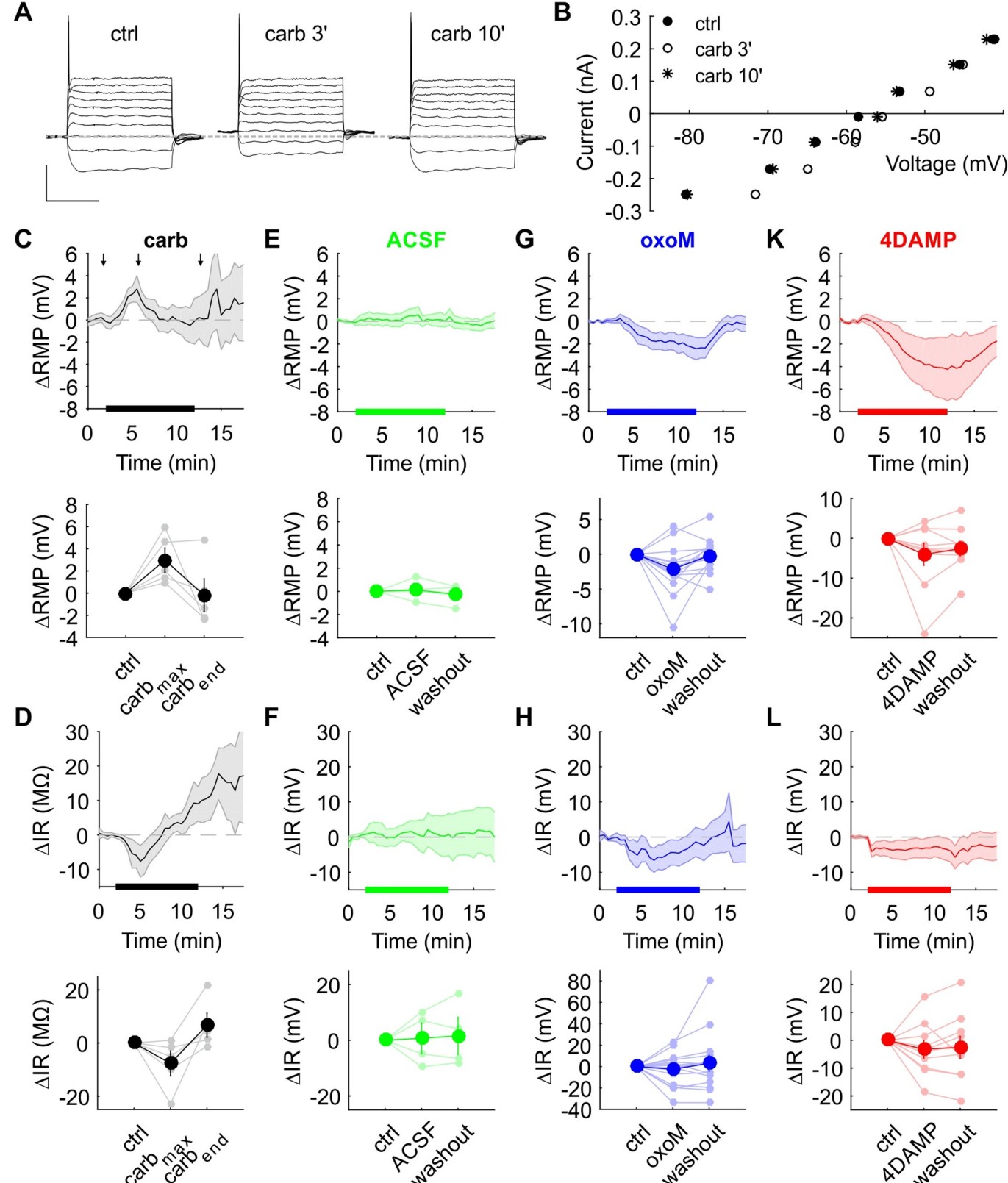

**Fig 9. RMP and IR monitoring during application of cholinergic compounds.** (A) Voltage response to 10 current step injections in control (left), after 3 min (middle) and 10 min (right) of carbachol perfusion; scale: 200 ms/30 mV. (B) Current-voltage curves of the recordings shown in (A). (C) Mean of RMP change

over time (upper part). The thicker black line on the X axis represents the duration of carbachol perfusion. The 3 arrows indicate the time of the recordings shown in A. The average of 2.5 min was plotted for each condition (lower part). (D) Mean of IR change over time (upper part) and average over 2.5 min in each condition (lower part). (E) Mean of RMP change over time (upper part). The thicker green line on the X axis represents the duration of the perfusion of ACSF (control experiment). (F) Mean of IR change over time (upper part) and average over 2.5 min in each condition (lower part). (G) Mean of RMP change over time (upper part). The thicker blue line on the X axis represents the duration of the oxoM perfusion. (H) Mean of IR change over time (upper part) and average over 2.5 min in each condition (lower part). (K) Mean of RMP change over time (upper part). The thicker red line on the X axis represents the duration of the 4DAMP perfusion. (L) Mean of IR change over time (upper part) and average over 2.5 min in each condition (lower part). Shaded areas represent the standard error.

(n = 2) or no (n = 3) response from 5 cells. No significant variation of the IR ($F_{(2,28)}$ = 0.68, p = 0.44, n = 15) could be measured for oxoM. The same scheme, though stronger (maximum change equal to -4.2 ± 1.5 mV), was visible with 4DAMP perfusion in Fig 9K (upper part). At the end of 4DAMP perfusion, the RMP increased slowly toward its initial value. Again, three cells showed opposite effects (depolarization) preventing statistical significance ($F_{(2,18)}$ = 2.25, p = 0.16, n = 10; lower part in Fig 9K). The effect of 4DAMP perfusion on IR was unclear (Fig 9L, upper part). The high variation between the cells and the lack of any recovery afterwards (Fig 9L, lower part) suggested an insignificant change ($F_{(2,18)}$ = 0.67, p = 0.46, n = 10).

We then analyzed two further functional parameters, namely the action potential threshold and the amplitude of the voltage sag during hyperpolarizing current injections in the same dataset (cf. Fig 9). This data is presented in Fig 10 in a similar format as Fig 9. Fig 10A shows current clamp recordings from a SBC before (Fig 10A, left), during 1 μM oxoM perfusion (Fig 10A, middle) and after recovery (Fig 10A, right). This SBC responded with strong hyperpolarization to the oxoM perfusion (RMP went from -60.1 ± 0.7 mV to -81.9 ± 1.0 mV and recovered to -63.9 ± 0.6 mV). The voltage sag, i.e. the difference between the maximum and the steady-state potential upon hyperpolarizing current injection, is changed during oxoM application. In this example cell the difference between peak- and steady-state hyperpolarization was 8.1 ± 0.2 mV during control, was reduced to 5.5 ± 0.2 mV during oxoM perfusion and recovered to 6.8 ± 0.5 mV after oxoM washout. This finding agrees with a reduction of $I_h$ during oxoM application (cf. Fig 4F). In Fig 10B we show an example of the AP threshold analysis. We identified the AP threshold by a rate threshold, i.e. the membrane potential where the rate of membrane potential change first exceeded 50 mV/ms. This is visualized for the control in Fig 10B. During the control the AP threshold was -37.7 ± 0.3 mV for this cell, was then reduced to -61.3 ± 0.7 mV during oxoM perfusion and recovered back to -41.8 ± 0.9 mV after oxoM washout. Similar to the data shown in Fig 9 we performed the measurements shown in Fig 10A and 10B every 30s for several minutes before, during and after perfusion of cholinergic agents. In Fig 10C & 10G we show the effect of carbachol perfusion on AP threshold and voltage sag. While AP threshold is only transiently reduced by carbachol immediately after wash-in (arrow in Fig 10C; reduction of -8.1 ± 7.3 mV, n = 5), carbachol causes a lasting reduction of the voltage sag (Fig 10G, on average reduction of -1.8 ± 1.9 mV). When tested with one-way ANOVA with repeated measures, these changes are statistically not significant though (AP threshold: $F_{(2,8)}$ = 0.07, p = 0.83, n = 5; voltage sag: $F_{(2,8)}$ = 2.9, p = 0.15, n = 5). Omitting the cholinergic agent (Fig 10D & 10H) neither caused a reduction in AP threshold ($F_{(2,6)}$ = 0.54, p = 0.53, n = 4) nor voltage sag ($F_{(2,6)}$ = 1.1, p = 0.37, n = 4), but showed how variable these metrics can unfortunately be.

Both washing in of oxotremorine-M (Fig 10E & 10J) and 4DAMP (Fig 10F & 10K) caused a slower and longer lasting reduction in AP threshold (oxoM: reduction of -3.4 ± 5.5 mV; 4DAMP: reduction of -4.3 ± 8.8 mV), but again due to the high variability of the data the results were not statistically significant (oxoM: $F_{(2,28)}$ = 1.1, p = 0.35, N = 15; 4DAMP: $F_{(2,18)}$ = 2.02, p = 0.17, n = 10). Surprisingly, washing in of oxotremorine-M did, averaged over all cells, only cause a small (non-significant) change in voltage sag (Fig 10J; $F_{(2,28)}$ = 0.99,

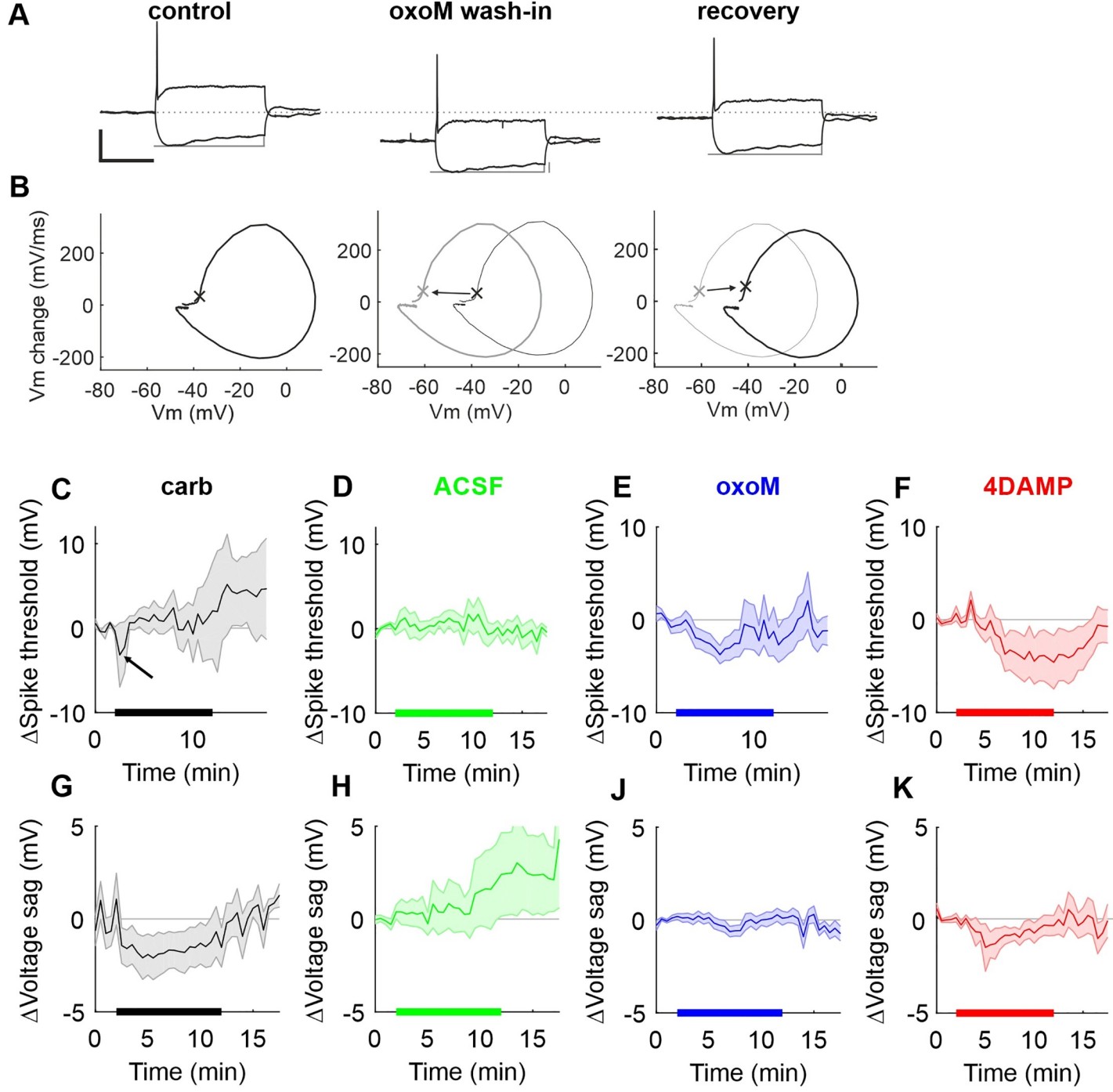

**Fig 10. AP threshold and voltage sag monitoring during application of cholinergic compounds.** (A) Voltage response to de- and hyperpolarizing current injections in control (left), during (middle; wash-in) oxotremorine-M perfusion and after (right; recovery) oxotremorine-M perfusion; scale: 50 ms/25 mV. Thin grey lines illustrate the difference between maximum- and steady-state depolarization, which we analyze here as voltage sag caused by activation of $I_h$. (B) Phaseplane plots illustrating the relation between the rate of change of the membrane potential and the membrane potential during the occurrence of the action potentials shown above. A value of >50 mV/ms was taken as AP threshold (marked with an x; note that due to the low sample number during the rapid rise the exact value of the threshold can be different in the three plots). It becomes clear that the AP threshold shifts towards hyperpolarized values during oxoM application (middle panel: thick grey line = oxoM, thin black line = control) and recovers after wash-out (right panel: thin grey line = oxoM, thick black line recovery). (C-F) Mean of AP threshold change over time for carbachol (C), ACSF (D), oxotremorine-M (E) and 4DAMP (F). The thicker line on the X axis represents the duration of perfusion. The arrow in C highlights the transient reduction of AP threshold caused by carbachol. (G-K) Mean of voltage sag change over time for carbachol (G), ACSF (H), oxotremorine-M (J) and 4DAMP (K). Shaded areas represent the standard error.

p = 0.37, n = 15). 4DAMP caused a transient reduction of the voltage sag (reduction of -1.7 ± 3.6 mV, Fig 10K), which was statistically not significant (F(2,18) = 0.8, p = 0.41, n = 10) due to high variability.

Although these data are hard to interpret due to high variability we can nevertheless conclude that a cholinergic, and specifically muscarinic, influence on both AP threshold and voltage sag appears likely for SBCs, which would corroborate our results from the voltage clamp experiments.

In conclusion, long-lasting perfusion of carbachol during current-clamp recordings from SBCs indeed allowed the visualization of depolarizing changes caused by the combined action of both nicotinic and muscarinic effects. Perfusion of M3 antagonists had a hyperpolarizing effect on the unclamped RMP, likely due to blocking of tonically activated muscarinic signaling (cf. [10]) contributing to the RMP. Surprisingly, the muscarinic agonist oxoM on average also hyperpolarized the RMP of SBCs. However, a considerable diversity regarding both direction of effects and temporal profiles of effects of muscarinic transmission in SBCs was evident.

## Discussion

We demonstrated that $I_M$ was present in SBCs. Both $I_M$ and $I_h$ were reduced by the perfusion of a muscarinic agonist. In addition, the presence of two muscarinic receptors M1 and M2 in the AVCN was confirmed by immunohistochemistry. M1 was somatic while M2 was mainly located in the neuropil and sparsely on the soma of SBCs. Although M2 is known to be localized on the presynaptic site, no effect was visible on the endbulb of Held to SBC synaptic transmission. Finally, we showed that muscarinic activation induced differential effects on the resting membrane potential of SBCs. The activation of all muscarinic receptors by oxoM on average hyperpolarized the RMP (-2.4 ± 1.0 mV, Fig 9G). The inhibition of M3 shifted the RMP to even more negative values (-4.2 ± 1.5 mV, Fig 9K).

$I_M$ is a voltage-dependent outward potassium current. When the membrane potential shifts to more positive values, M channels slowly open and thereby hyperpolarize the membrane potential and bring the cell closer to its resting state. The M-channels consist of a heteromeric assembly of Kv7.2 and Kv7.3 potassium channel subunits [32,33]. The presence of low-threshold potassium conductance in SBCs, which could correspond to M conductance, has been already reported by [34]. In this study we confirmed that $I_M$ is present in SBCs. Indeed, the perfusion of the Kv7 blocker XE991 almost completely abolished the inward relaxation current. The activation of all muscarinic receptors via local application of oxoM also abolished the inward relaxation (Fig 3). In addition, we showed that long application of M3 muscarinic receptor antagonist (4DAMP) induced a negative shift of SBC resting membrane potential (Fig 9K) similar to the change caused by tolterodine (broad muscarinic antagonist) bath application and confirming a tonic activation of muscarinic receptors [10]. Immunohistochemical data confirmed the presence of M3 on the soma of SBCs [5]. These results indicate that M3 receptor activation likely causes the long-lasting depolarization through the closure of M-channels. Several studies have demonstrated that another muscarinic receptor, M1, was also involved in $I_M$ inhibition [28,35,36]. Our immunohistochemical staining exhibited strong M1 expression on the soma of AVCN cells including SBCs (Fig 6A–6D). M1 and M3 are both Gq/11 protein-coupled metabotropic receptors triggering the same intracellular signaling (Fig 11). Acetylcholine (or any other cholinergic agonist) binding to M1/M3 leads to the cleavage of the phosphatidylinositol-4,5-biphosphate (PIP$_2$) into diacylglycerol (DAG) and inositol triphosphate (IP$_3$) through activation of phospholipase-Cβ (PLCβ). However, the exact secondary messenger which directly induces the closure of M-channels is still debated.

A noteworthy result is that XE991 failed to completely abolish the inward relaxation current at the most depolarized values tested (Fig 3A) while oxoM inhibits the current at all voltages. This could be explained by the difference in the application system of the drugs. OxoM was locally applied via a perfusion pencil just above the area where the cell was located while XE991 was perfused in the bath chamber. The former could allow a faster activation of muscarinic receptors whereas the replacement of normal ACSF by ACSF containing XE991 took several minutes. However, this was compensated by waiting at least 10 min of XE991 bath application before recordings, leaving enough time to totally replace the ACSF in the bath chamber and affect all cells. An alternative explanation is that Kv7 channels are not the only ion channel type involved in $I_M$. Indeed, only an HCN blocker was added to the ACSF in these experiments. The remaining current had an activation threshold more positive than $I_M$ (starting around -40 mV) which matches with the delayed-rectifier potassium current ($I_{KDR}$). Although $I_{KDR}$ has faster kinetics than those of $I_M$, our measurements of the inward relaxation did not consider the kinetics and consequently a role of $I_{KDR}$ in this cannot be excluded. Other channels could also be involved in $I_M$. For example, two-pore-domain potassium channels were closed by the activation of M1/M3 muscarinic receptors causing the thalamo-cortical neurons to depolarize [15].

$I_h$ is a depolarizing cationic current, mainly due to an influx of sodium. The molecular correlate is the hyperpolarization-activated cyclic nucleotide-gated (HCN) channel. The name $I_h$ comes from its unusual voltage-dependency. Indeed most of the channels are open when the membrane potential is hyperpolarized but a proportion of HCN channels stay tonically open at resting potential and maintain a resting state close to action potential threshold [20,37]. Four different isoforms of HCN channels exist, HCN1 to HCN4. Mouse BCs strongly expressed the isoforms HCN1, moderately HCN2 and HCN4 [16]. The local application of oxoM strongly reduced $I_h$ at every voltage tested (Fig 5) meaning that muscarinic activation here inhibits a depolarizing current, thereby hyperpolarizing the cell membrane. Part of this effect can result from the decrease of adenosine-3',5'- cyclic monophosphate (cAMP) and/or guanosine-3',5'- cyclic monophosphate (cGMP). These cyclic nucleotides regulate HCN channels by shifting the voltage of half-maximal activation ($V_{1/2}$) to more positive values [20,37]. Thus, reducing the cyclic nucleotide availability shifts the activation curve to a more negative value. In that case, the probability that HCN channels are open is reduced for the same voltage. Two muscarinic receptors, M2 and M4, induce a decrease of cAMP [31] by inhibiting the adenylate cyclase (AC). This enzyme is inhibited by the α-subunit of Gi/o protein, itself activated by M2/M4 receptors (Fig 11).

Our immunohistochemical data revealed the presence of M2 muscarinic receptors in the AVCN (Fig 6E–6H). The M2-positive signal was mostly detected in the neuropil of BCs, implying either a dendritic localization on SBCs or an expression on the presynaptic site. However, no overlap was observed between calretinin-positive auditory nerve terminals and M2 staining. In addition, presynaptic parameters, estimated by recording evoked and spontaneous EPSC, were unchanged upon wash-in of carbachol (Figs 7 and 8). However, if a tonic activation of muscarinic receptors occurs, as previously suggested by [10], a small effect might be more easily revealed by the application of an antagonist. Nonetheless, the absence of cholinergic effect on the synaptic transmission are in accordance with our immunohistochemical results. On the other hand, M2 localization on calretinin-negative terminals (possibly of cholinergic axon coming from the olivary complex or from the ponto-mesencephalic tegmentum) could not be refuted based on our results. Indeed, M2 is known to be an autoreceptor that can regulate acetylcholine release [31]. Furthermore we cannot exclude presynaptic localization on inhibitory terminals, which strongly influence responses of SBC. M2 staining was also localized on the soma of SBCs (Fig 6H, empty arrowheads) confirming that M2 was present on the

postsynaptic side in SBCs. The existence of M2 on the soma and dendrites of postsynaptic cells has previously also been demonstrated in hippocampal interneurons [38].

Recently, [21] have demonstrated that cholinergic interneurons expressing M2 exhibited a decrease of around 20% of $I_h$ under application of oxoM. However, the strong inhibition of $I_h$ observed here is unlikely to be explained only by the M2-mediated decrease in cAMP alone. This suggests that other phenomena influence HCN channels. There was evidence that another second messenger, PIP$_2$, affects the activation curve of HCN channels [39]. In that study, adding PIP$_2$ shifted the activation of HCN2 channels to positive voltages similarly to cAMP. Even more compelling, adding both cAMP and PIP$_2$ shifted $V_{1/2}$ to even more positive voltages than cAMP or PIP$_2$ alone. M1/M3 receptors expressed by SBCs then could indirectly be responsible for a significant part of $I_h$ inhibition (Fig 11, grey dashed arrow) through the cleavage of PIP$_2$. Furthermore, HCN channels are also regulated by their phosphorylation status. The protein kinase A (PKA), activated by cAMP, positively shifted the activation curve of $I_h$ (HCN4) isoform in mouse sinoatrial node [40] while the protein kinase C, activated by PLC, phosphorylated HCN1 and reduced $I_h$ in hippocampal neurons [41]. Both HCN channel isoforms are present in BCs [16]. In summary, a combination of possible intracellular mechanisms through which muscarinic activation can modulate $I_h$ seems likely and could explain the strong effects seen in our recordings.

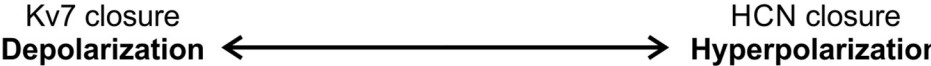

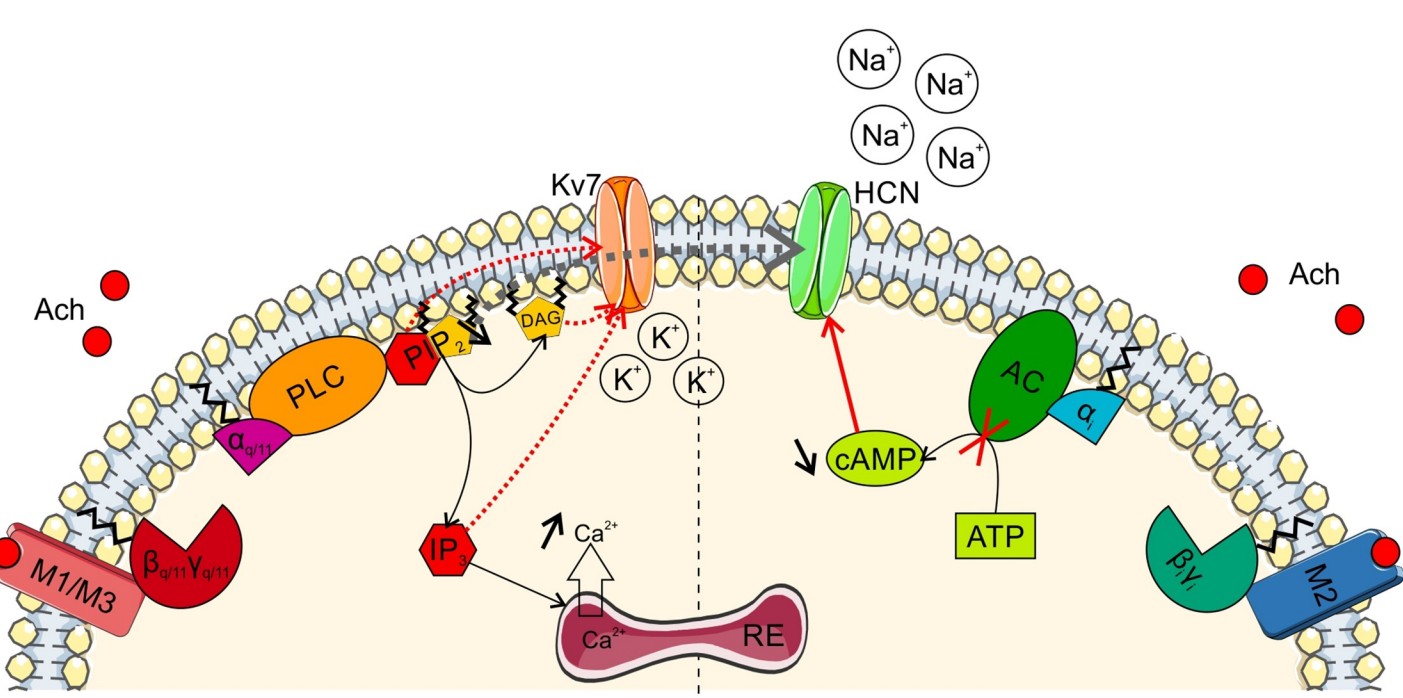

**Fig 11. Hypothetical signaling cascade in response to acetylcholine release.** On one hand, acetylcholine (Ach) binds the M1/M3 muscarinic receptor which induces the activation of the phospholipase C (PLC) via the G$\alpha_{q/11}$ protein. The activated PLC depletes the phosphatidylinositol biphosphate (PIP$_2$) into diacylglycerol (DAG) and inositol triphosphate (IP$_3$). IP$_3$ concentration increase releases calcium from the internal cell storage. Directly or indirectly, one or several of these 2$^{nd}$ messengers trigger the closure of Kv7 channels thereby depolarizing the cell membrane. On the other hand, acetylcholine may bind the M2 muscarinic receptor inhibiting the adenylate cyclase (AC) via the G$\alpha_i$ protein which induces a reduction of cyclic AMP. Thus, for the same membrane potential, more HCN channels are closed inducing the hyperpolarization of the cell. Cell membrane and ion channels were designed by "Servier Medical Art" (https://smart.servier.com) provided by Les Laboratoires Servier (https://servier.com/en), licensed under Creative Commons Attribution 3.0 Unported License.

Muscarinic activation in SBCs induced two opposite effects. M1/M3 activation blocked the M-current which should depolarize the RMP while M2 activation reduced $I_h$ and then supposedly leads to a hyperpolarization of the cell membrane. This result raises one important question: why does the same signal induce two opposite effects? The changes of the resting membrane potential, the input resistance and the action potential threshold were monitored under application of carbachol, oxoM or 4DAMP to clarify the resulting effect of the global muscarinic activation on the excitability of the neuron. Long application of carbachol activated both nicotinic and all types of muscarinic receptors. The RMP depolarized in the first three minutes after application of carbachol and went back to its initial value before washing out carbachol. Action potential threshold briefly decreased as well which, together with the depolarization, should strongly but transiently increase the spike probability of the SBC. This spontaneous return to the initial value before the end of carbachol application was previously also noticed *in vivo* in SBC spontaneous firing rate under iontophoretic application of carbachol [10]. Additionally, the transient depolarization was accompanied by a transient decrease in the input resistance consistent with the opening of nicotinic receptors generating a sodium inward current. The desensitization of nicotinic receptors can explain the spontaneous return of IR and RMP to their normal values.

Noteworthy, the input resistance continued to slowly increase even after the end of carbachol application. This high increase in the input resistance (almost 20 MΩ at 15 min, Fig 9D, upper part) could reflect the closure of both Kv7 and HCN channels though no concomitant significant change was observed in the RMP. This contradicts the long-lasting depolarization observed after carbachol puff application shown in [10]. Nevertheless, the RMP was unstable during the IR increase. One hypothesis is that the hyperpolarizing effect of $I_h$ reduction counteracts the depolarizing effect of $I_M$ blockade to stabilize the cell potential. The differential results between the two studies could be explained by the different system of carbachol application. The puff applications of carbachol used in [10] were phasic and very localized on the dendrites and part of the soma close to the primary dendrite while the perfusion pencil application of carbachol used in this study was sustained and reached every cell in a local volume of the slice and all the cellular compartments (whole dendritic tree, soma and initial axon segment). The long-lasting application can lead to the activation of a greater absolute numbers and a different ensemble of muscarinic receptors and might trigger their desensitization/internalization inducing a different overall change.

Surprisingly, long application of oxoM showed a distinct response. Naturally, the transient nicotinic effect was absent since only muscarinic receptors were activated. Nonetheless, the average RMP clearly hyperpolarized in presence of oxoM (Fig 9G, upper part) without a clear effect on the IR. The lack of effect on the IR might be due to big inter-individual variation. We also cannot rule out potential effects of high or fluctuating series resistance during the long recordings (cf. "Electrophysiology" in the Material & Method section). The hyperpolarization upon oxoM application could result from a disequilibrium between $I_h$ and $I_M$ in favor of $I_h$ inhibition. This can be caused by more activated M2 receptors than both M1/M3 receptors. Yet, our immunohistochemical data revealed a stronger M1 than M2 expression (Fig 6D and 6H respectively) which might be in conflict with this hypothesis. However, it seems that oxoM binds more selectively to M2 than M1 in comparison to carbachol [42] favoring $I_h$ inhibition causing the hyperpolarization. The 4DAMP perfusion caused a strong hyperpolarization. The blockade of M3 activation against the background of a tonic cholinergic signaling in SBC [10] leads to 1) $I_M$ inhibition only caused by M1 activation and 2) $I_h$ inhibition via M2 activation alone, resulting in a net negative shift of RMP.

Noticeably, few cells in both oxoM (n = 2) and 4DAMP (n = 3) perfusion experiments showed an opposite effect to the average. This can be explained by an imbalance in $I_h$ and $I_M$

inhibition depending on the proportion of M1/M3 vs. M2 expression. Indeed M1 seemed to be strongly expressed on all AVCN cells while M2 expression was sparse and heterogeneous (Fig 6). These results suggest an underlying diversity of cholinergic signaling in SBC based on factors not yet identified.

The variability observed in response to muscarinic activation might be explained by the range of gerbil age we chose. Indeed, all electrophysiological recordings were made in hearing gerbils aged between P14 to P25. However, the expression levels of both KCNQ and HCN are age-dependent in central neurons [43]. In our results, while the currents $I_h$ were larger in older animals, no age-related difference could be observed for M currents. In addition, no correlation could be made between the age and the diversity of RMP responses consecutive to the application of muscarinic modulators suggesting that the diversity observed in SBCs is unlikely age-dependent.

In general, the muscarinic effect observed on the currents and on the RMP in the different whole-cell recordings could have been underestimated. Indeed, we proposed a model in which secondary messengers (like cAMP, IP$_3$) of muscarinic activation act on the channels but they might have been diluted/washed out over time. This might also have contributed to the variability in the results. Furthermore the artificial extracellular calcium concentration and the reduced recording temperature might exert differential effects on different second messenger pathways. We would thus suggest to perform perforated-patch recordings at physiological temperature and ion concentrations as a control in future experiment to rule these possible problems out.

In this study, muscarinic receptors were activated by the application of oxoM, but this agonist has two downsides that could affect the interpretation of our results. First, we argued that oxoM inhibited M currents by blocking Kv7 (KCNQ) channels via the activation of muscarinic receptors (see Figs 2 and 3). However, it has been demonstrated that oxoM can also directly block KCNQ2/3 channels [27]. If this direct mechanism on Kv7 channels occurred here, we could not distinguish whether the inhibition of M currents resulted from muscarinic activation or from a direct blockade of KCNQ2/3 channels by oxoM. However, Kv7 blockade should logically lead to cell depolarization; but, on average, the cell RMP hyperpolarized (see Fig 9) upon oxoM application. We suggested that this hyperpolarization resulted from $I_h$ inhibition consecutive to M2 activation. This effect cannot be explained by the direct effect of oxoM on KCNQ2/3. In addition, a muscarinic antagonist tolterodine shifted the cell RMP to more negative values [10] which supports the hypothesis that muscarinic activation depolarizes the cell, supposedly by blockade of Kv7 channels. Secondly, it was shown that oxoM directly and indirectly potentiates NMDA currents [27]. However in mature AVCN the transmission is mostly mediated by AMPA receptors [44]. Thus this effect likely does not play a large role here.

One hypothesis to explain the purpose of having two opposite effects (depolarization vs. hyperpolarization) triggered by the same molecule is the following. In the case of a brief application (like puff application), carbachol binds with a higher probability M1/M3 receptors (due to higher expression) resulting in a depolarization of the resting membrane potential. During long lasting bath application, carbachol reached every cell part leading to more M2 activation which can counteract the effect of M1/M3 depolarization and bring the RMP back to its initial value. This phenomenon could represent a form of adaptation in case of a strong and long-lasting cholinergic activation and could also explain the spontaneous return of the normal spontaneous firing rate of SBCs in the in vivo experiments under high concentration of carbachol [10]. As described in [10], the transient cholinergic excitation (RMP depolarization) likely enhances sound localization by providing more well-timed action potentials and seems to improve the detection of tones in noisy background by expanding the dynamic range of responses. Depending on the exact kinetics of the partially opposing effects of cholinergic

transmission on SBC we demonstrated, one could speculate about the listening situations each would have the greatest impact in. In listening situations rich in transients, with rapidly changing stimulus levels, the depolarizing effects of cholinergic transmission could dominate. This would render SBC more excitable and counteract peripheral cholinergic gain control as discussed in [10]. However, during long durations of intense stimulation (e.g. constantly loud surroundings) the hyperpolarizing muscarinic effects we saw in our wash-in experiments could dominate. This might act as an additional central gain control that reduces tonic activation (and thus masking effects) of SBC by the background noise. Direct physiological evidence for these ideas is not yet available unfortunately.

In conclusion, both $I_M$ and $I_h$ are present in SBCs and are both modulated by the activation of muscarinic receptors. $I_M$ inhibition is most likely mediated through the activation of both M1 and M3 receptors whereas $I_h$ is most likely decreased by M2 activation. The resting membrane potential might be tuned by the balance of these currents and then depends on the relative proportion of M1/M3 and M2 receptor activation. Further experiments are needed to understand the intracellular signaling caused by M1/M3 and M2 activation and the physiological meaning of two opposite effects induced by the same modulator in the same cell.

## Author Contributions

**Conceptualization:** Charlène Gillet, Thomas Kuenzel.

**Data curation:** Charlène Gillet, Thomas Kuenzel.

**Formal analysis:** Charlène Gillet, Thomas Kuenzel.

**Funding acquisition:** Thomas Kuenzel.

**Investigation:** Charlène Gillet, Stefanie Kurth.

**Methodology:** Charlène Gillet, Stefanie Kurth, Thomas Kuenzel.

**Project administration:** Charlène Gillet, Thomas Kuenzel.

**Resources:** Charlène Gillet, Stefanie Kurth.

**Software:** Charlène Gillet, Thomas Kuenzel.

**Supervision:** Thomas Kuenzel.

**Validation:** Charlène Gillet, Thomas Kuenzel.

**Visualization:** Charlène Gillet.

**Writing – original draft:** Charlène Gillet.

**Writing – review & editing:** Charlène Gillet, Thomas Kuenzel.

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
