## [Decision Letter · Decision Letter 0]

24 Jul 2019

PONE-D-19-18119

Muscarinic modulation of M and h currents in gerbil spherical bushy cells

PLOS ONE

Dear Dr Kuenzel,

Thank you for submitting your manuscript to PLOS ONE. After careful consideration, we feel that it has merit but does not fully meet PLOS ONE’s publication criteria as it currently stands. Therefore, we invite you to submit a revised version of the manuscript that addresses the points raised during the review process.

Reviewers and the editor found the study interesting, however there are issue that require to be addressed before the paper can be considered suitable for publication. Please address all the concern raised during the revision.

We would appreciate receiving your revised manuscript by September 30th 2019. To enhance the reproducibility of your results, we recommend that if applicable you deposit your laboratory protocols in protocols.io, where a protocol can be assigned its own identifier (DOI) such that it can be cited independently in the future. For instructions see: http://journals.plos.org/plosone/s/submission-guidelines#loc-laboratory-protocols

We look forward to receiving your revised manuscript.

Kind regards,

Andrea Barbuti, PhD

Academic Editor

PLOS ONE

Journal Requirements:

1. We note that you have stated that you will provide repository information for your data at acceptance. Should your manuscript be accepted for publication, we will hold it until you provide the relevant accession numbers or DOIs necessary to access your data. If you wish to make changes to your Data Availability statement, please describe these changes in your cover letter and we will update your Data Availability statement to reflect the information you provide.

Reviewers' comments:

Reviewer's Responses to Questions

**Comments to the Author**

1. Is the manuscript technically sound, and do the data support the conclusions?

Reviewer #1: Yes

Reviewer #2: Yes

2. Has the statistical analysis been performed appropriately and rigorously? 

Reviewer #1: Yes

Reviewer #2: I Don't Know

3. Have the authors made all data underlying the findings in their manuscript fully available?

Reviewer #1: Yes

Reviewer #2: Yes

4. Is the manuscript presented in an intelligible fashion and written in standard English?

Reviewer #1: Yes

Reviewer #2: No

5. Review Comments to the Author

Reviewer #1: The manuscript by Gillet, Kurth and Kuenzel investigates the presence and effect of muscarinic receptors on spherical bushy cells in the anteroventral cochlear nucleus. The authors use a combination of pharmacology and immunohistochemistry to identify the presence of M1 and M2 receptors on those cells which are differentially expressed on the cell soma and neuropil, respectively. Using agonists for muscarinic receptors (Oxotremorine M) and antagonist for Kv7 (XE991) and HCN channels (ZD7288), the authors show that the effect of mACh activation largely overlaps with block of Kv7 and HCN channels and conclude that muscarinic receptors in SBCs interact with Kv7 and HCN channels resulting in long lasting changes of resting membrane potential.

Finally, the authors propose a model in which the endogenous release of acetylcholine acts on both M1/M3 and M2 receptors. Activation of M1/M3 receptors will depolarize the cell by closure of Kv7 channels while activation of M2 receptors hyperpolarizes the cell by closure of HCN channels.

Overall, the manuscript provides a detailed investigation of the role of muscarinic acetylcholine receptors on SBC and is in line with previous reports by the same authors, suggesting an intriguing role of cholinergic modulation on SBCs. The data is solid and the interpretation and conclusions are mostly appropriate and close to the data.

However, the use of pharmacological agents to dissect metabolic pathways has its inherent caveats and I feel that some points need further clarification to strengthen the author’s interpretation.

Major comments:

1. The expression of M1/M2 receptors on SBCs has been investigated by immunohistochemistry and important conclusions are based on those data. However, the staining against M1 suggests a predominantly cytoplasmic localization, rather than membrane bound, which might be caused by unspecific binding of the antibody. Did the authors confirm the specificity of the antibody or performed the control experiment using an antibody that bounds another epitope (e.g. Alomone Labs Cat# AMR-001, RRID:AB_2039993 binds the third intracellular loop, rather than the C-terminus of M1 receptors)?

2. The series resistance during ephys recordings was not compensated, but the authors do not report Rs values during the recordings or if there were any differences between groups or changes over time. While the influence on the slow and rather small IM currents might be negligible, a high Rs will likely affect the fast, larger IInst currents which in turn could affect the analysis of IM. Likewise, the amplitude and kinetics of AP-evoked EPSCs (Figure 7) will critically depend on the recording’s Rs due to the non-linearity of Rs compensation. Unless the uncompensated Rs values during recordings were either small (<5 MOhm) or very similar between groups this could introduce considerable errors when estimating amplitude and kinetics. The authors should therefore also report the Rs values for the individual recordings or declare if recordings with high Rs (e.g >>10 MOhm) were discarded.

3. The authors recorded from P14-P30 gerbils of either sex. In their previous study (Gillet, 2018, J. Comp. Neurol.), the authors reported an increase in cholinergic innervation between P15 and P28 and another recent study (Mueller et al, 2019, Front. Cell. Neurosci., DOI: 10.3389/fncel.2019.00119) suggest that some properties of mouse CN neurons develop after hearing onset. It is not clear why the age range was chosen so close to the animal’s hearing onset and no information is provided on the age distribution for the different experiments (except for immunohistochemistry at P30). Considering the contrasting effects of oxoM and 4DAMP in some of the recordings (Figure 9, some cells show increase in RMP, other decrease), did the authors split the data set by age or sex to determine if those factors might account for the observed variability, e.g. activation of mACh might be depolarizing in young, but hyperpolarizing in older animals or vice versa. Since the authors have those data readily acquired, it might be worth adding a statement, if any difference in sex/age was observed. If a certain age range was used for some of the recordings for experimental reasons (e.g. recordings more stable), this should be mentioned.

4. Based on fiber stimulation and mEPSC recordings with application of carbachol, the authors conclude that cholinergic transmission does not affect synaptic transmission at SBCs. However, the authors also show that application of 4DAMP (an M3 antagonist) hyperpolarizes SBCs, suggesting a tonic activation of M3 receptors in the slice. If ACh receptors were tonically activated at the endbulb of Held, the effect on synaptic transmission would likely be negligible when using ACh agonists but would require the use of ACh antagonists. This possibility should be tested or at least discussed.

5. The authors performed fiber stimulation at 100 Hz and analyzed the EPSC kinetics for the first EPSC. Did the authors also test higher (e.g. 300 Hz) stimulation rates, which would more closely mimic in vivo activity? A potential ACh modulation of synaptic vesicle release might be more pronounced during ongoing activity. Similarly, the authors analyzed EPSC kinetics for the first EPSC only. Since the first EPSC after a period of quiescence is somewhat artificial (considering the high spontaneous rates of most auditory brainstem neurons), it might be worthwhile to also analyze the EPSCs later in the train (if the signal-to-noise ratio permits such analysis).

6. Application of the mACh receptor agonist oxoM hyperpolarized the SBC and the authors argue that an indirect block of Kv7 channels causes this effect. While this argument is conceivable and has been shown in other systems, the authors should be aware that oxoM can directly block Kv7 channels without the activation of mACh receptors (see Zwart et al., 2016, Eur. J. Pharm. DOI: 10.1016/j.ejphar.2016.08.037). Did the authors perform control experiments with oxoM + atropine to block mACh receptors or used a more specific mACh agonist (e.g. oxotremorine or xanomeline) which has been shown to not interact with KCNQ channels? If not, the authors should at least discuss this possibility and how it might affect their conclusions.

7. The model in Figure 10 proposes two opposing effects of endogenous ACh, one leading to depolarization by Kv7 closure, the other to hyperpolarization by HCN closure. The authors speculate that brief ACh application will predominantly bind to M1/M3 receptors (highly expressed), thereby depolarizing the cell, while during prolonged application the hyperpolarization by M2 (weakly expressed and located at the neuropil) activation dominates. While this explanation is reasonable in the current experimental setting, it might not apply in vivo (since the origin and activation dynamics of those inputs are barely known). The authors should discuss if either of those conditions might be favorable during varying acoustic stimulation (e.g. hearing in noise, natural sounds).

8. The proposed model suggests the involvement of secondary messengers (cAMP, PLC etc) to act on Kv7 and HCN channels. While this is reasonable and has been shown in other systems, I wonder if some of the reagents have been washed out/diluted over time in the whole-cell experiments, thus underestimating the observed effect. This could be tested by e.g. gramicidin-perforated patch-clamp experiments. Do the authors have any data on this? If not, new experiments are not required, but the authors should acknowledge and discuss this possibility.

9. Line 502: “oxoM induces closure of M channels […] and depolarizes the resting membrane potential”. This conclusion is not immediately evident from the data, since application of oxoM (Figure 9G) resulted in an, on average, small hyperpolarization. The authors might argue that block of M3 channels hyperpolarized the cell, but unless M1/M3 channels were specifically activated and result in depolarization, this statement is too strong given the underlying data.

10. Previous studies by the corresponding author and others (e.g. PMIDs: 21411667, 25164657, 28945194, 27855778, 23345233) point to an important role of inhibition in shaping SBC responses, particularly during complex or naturalistic acoustic stimulation. Did the authors find any evidence for ACh receptors on inhibitory inputs contacting the SBCs?

Minor comments:

11. Slices were incubated at room temperature and all recordings were performed at room temperature and at supra-physiological calcium (2 mM vs. 1.2-1.5 mM). Since the kinetics of G-protein mediated signal transduction pathways are temperature and calcium dependent, the observed effect in the experiment might be different to in vivo conditions, especially considering the proposed “balance” between M1/M3 (depolarizing) and M2 (hyperpolarizing) mediated pathways. While I acknowledge that the authors performed similar experiments at 37C in a previous publication (Figure 4 in Goyer et al, 2016, eNeuro), it might be worth discussing the effect of temperature and calcium concentration in the light of the two proposed pathways.

12. The manuscript would benefit from a statistics section within the Methods. Although the statistics for this data set is mostly straight forward, the authors should describe how sample size and statistical tests were selected (e.g. testing for normal distribution, etc.).

13. Line 98: Please state how the junction potential was estimated, by calculation or experimentally. Was the junction potential also corrected for the voltages shown in Figure 9B?

14. Line 156: Please state the concentration of Triton X-100 in PBS. Was it 0.1 % as in the subsequent steps?

15. Line 79: Please indicate which temperature was considered room temperature (e.g. 25C).

16. For antibodies and software used, please include RRIDs where available (e.g. line 192 “rabbit anti-M1 muscarinic receptor (Alomone Labs Cat# AMR-010, RRID:AB_2340994)”)

17. In the Methods, the authors describe that cells were filled with biocytin during recordings, but its purpose appears unclear, since the immunohistochemistry was performed on animals not used during slice recordings. If the biocytin staining was used to morphologically verify the cell type after the end of recordings (e.g. to identify SBC vs. stellate cells), it should be stated in the Methods and Results section.

18. Line 143: Please indicate how many orders were used for the polynomial fit and if any constraints were used.

19. The manuscript frequently reports p-values as p=1. While I understand that p-values may be >1 after applying the Bonferroni correction on the actual p-values rather than the alpha value of the test, it is better to report such p-values as p>0.9999, since p=1 would only be possible if the means of two groups are identical.

20. Figure 1: In the bottom part of Panel A, the response to multiple depolarization steps is shown, but the indication of IM, ISS etc. is for one of the traces only, which might be confusing. Consider indicating for which trace the labels apply, e.g. using different colors or line styles.

21. Figure 1: Consider replacing the title of Panel C “IM at -50mV” with “Determination/Identification of IM at -50mV” or alike, since the panel shows all currents, not only IM

22. Figure 2: The panel organization is hard to follow. Consider moving the current-voltage graphs next to the corresponding traces. Also, it appears that the same data is labeled “XE991” in panel D and “ctrl” in panel G. While I understand the rationale of doing so, it might be clearer to combine those two panels into a single graph (labeled as ctr, XE991, XE991+oxoM which is then consistent with the raw traces). It might also be easier to follow, if the y-labels of panels D/F/G were labeled “Inward relaxation current IM”, as to link it to IM in panel C.

23. Figure 2: The y-labels on panels D/F/G is not centered on the y-axis

24. Figures 3-5: Please be consistent regarding the nomenclature of the currents. For instance, in Figure 3A-C the current is labeled “inward relaxation current”, but labeled “IM” in panel D. In my understanding those are the same currents but using different names is potentially confusing. Similar in Figures 4/5 with most panels labeled “inward current”, but “inhibition of Ih” in Fig. 5D. In my understanding, the term “inward current” is incorrect in this context, since it only refers to Ih, not the total inward current (as indicated in Figure 4C)

25. Figure 3 & 5: The currents at -50 mV (Figure 3) and -110 mV (Figure 5) were analyzed using paired t-tests. Since multiple measures have been taken from the same cells (ctr vs. drug and at different voltages) I feel that a two-way repeated measures ANOVA would be more appropriate and informative. In this case, the data should be analyzed using the within-subject factors treatment (ctrl vs. drug) and voltage.

26. Figure 6: The DAPI labeling looks unusual and different from previous papers from the same authors (e.g. Gillet et al, 2018, J. Comp. Neurol.), i.e. nuclei are barely labeled (see the “holes” in the cells in panel D, which should contain the cell nuclei) and most of the labeling is constrained to the auditory nerve root. Since the DAPI labeling is not critical for the conclusion (and not referred to in the manuscript), consider removing this channel from the images.

27. Figure 6: Please include which magnification (e.g. 60x) was used for imaging.

28. Figure 8: Since the detection of mEPSCs is highly variable and critically dependent on threshold selection, it would be helpful to indicate detected mEPSCs in the representative traces shown in panel A, so as the reader has a better understanding of which events were considered mEPSCs

29. Figure 8 & lines 395-404: The data are analyzed using ANOVA, but not details are provided as to which type of ANOVA was used. Given the experimental design, it seems a one-way repeated measures ANOVA would be appropriate. Please indicate this in the text.

30. Figure 9: Please rescale the y-axis in panel H bottom to avoid cutting off data points.

31. Figure 10: Please label the small red circles as “Acetylcholine”.

32. Line 223: “Array of blockers” awkward wording, perhaps replace with “combination/cocktail of blockers”

33. Line 360: consider replacing “bigger M2 signals” with “intense M2 signals”

34. The authors should consider replacing the bar charts in Fig. 7/8 with box plots but keeping the individual data points to better display the data distribution.

35. Lines 465-475: The authors explained that when excluding parts of the data set (cells with no or depolarizing effect) the statistical tests became significant. While this is not surprising (even a normal distribution with enough N will show “statistical significance” when limiting the data set to only positive or negative values), it implies that those excluded cells “prevented” statistical significance. Unless those cells were excluded based on criteria (e.g. age, sex, recording quality etc.) other than their response to drug application, this exclusion seems unjustified and potentially misleading. The authors did a good job in pointing out the differences in responses but should probably remove the statistics of the partial data set (after exclusion of those cells).

36. Line 498: “For the first time […] IM was shown in SBCs”. This statement is too strong considering that Marx & Manis, 1991 (J Neurosci, DOI: 10.1523/jneurosci.11-09-02865.1991) reported a very similar low-threshold conductance and originally discussed the presence of an M current in bushy cells.

37. Line 515-416: unclear what is meant by “other synaptic functions”? The authors did not test the effect of cholinergic transmission on e.g. LTP or homeostatic plasticity, so this statement is vague and should be rephrased.

38. Lines 26 & 564: Consider removing/replacing the subjective term “interestingly”.

39. Line 143: typo in “substracting” (remove second “s”)

40. Line 149: replace “glycinergic and GABAergic blockers” with “glycine and GABA receptor blockers”.

41. Line 374: replace “immunohistocheminal” with “immunohistochemical”

42. Line 424: “cell conductance” should probably read “cell capacitance”.

43. Line 430: replace “showed in (A)” with “shown in (A)”

44. Line 569: missing “t” in “spontaneous”

45. Line 539: please explain what “nACSF” stands for, if the additional “n” is not a typo.

46. Line 540: replace “a HCN blocker” with “an HCN blocker”.

Reviewer #2: The muscarinic modulation of M and H currents in gerbil spherical bushy cells by Gillet and colleagues has potential to be of interest to the wide audience of PLOSONE, but it requires a thorough revision to improve its clarity.

Major issues

The manuscript needs some grammar and language polishing. Some sentences seem incomplete or awkward. For instance: pg 12, ln 273 “in control conditions of by the …”.

In some cases (Fig. 2 legend), the description of the data is too succinct.

Although the authors do explain that they use whole cell recordings to identify the muscarinic receptor subtypes and the currents involved in spherical bushy cells, and this is indeed very interesting an sound, the manuscript would benefit from a clearly stated hypothesis.

The age range of the animals is 14-30 days. While 14 days is juvenile, 30 days is considered as young adult, which may lead to increased variability of the results. Also, the expression levels of KCNQ and HCN channels is age dependent (see (Buskila et al., 2019) and should be discussed.

It is unclear which protocol was used to measure Ih? How can it be the same as for Im (P6)? Please explain

Please explain why you chose the concentration of Carbachol, as 500uM seems a bit high.

Fig 1. I’m not aware about all the plethora of K+ channels expressed in BSC, but the fact that one can see an outward current not necessarily imply that this current is Im. The authors should add a selective blocker to confirm the identity of this current, as done in fig 2.

Also, I’m not sure why fig. 1 and fig. 2 are not joined. I don’t see any added value for both figures

Fig. 3d – is not clear. Please re-label the graph to clearly identify the difference between XE991ctrl and XE991, OxoM-xe991 and OxoM ctrl.

Fig. 9E legend– perfusion of what? nACSF? What is it, not explained in the text.

Fig. 9G – the current data show no significant difference if all data is included and becoming significant if some data is excluded. If there is a good reason for excluding data, please do so. If the authors convinced that there is a trend, they should increase the n-number.

Statistical tests – It is not clear which ANOVA was used? One way, two way? What was the significance level and df? These should be clearly stated.

Oxotremorine-M potentiate NMDA currents; however, the authors seems to neglect this. How can it affect the results? Ca2+ and Na+ entry?

Immunohistochemistry data– how many slices were examined? No stat is reported.

Although the authors looked into the impact of Im and Ih on the subthreshold activity in BSC (RMP, IR), the suprathreshold physiological activity (firing rate, resonance, SFC’s) was not tested. If the authors have the data, they should present it as it would strengthen this study findings.

Minor issues

Pg 3, Ln 39 – please identify VCN as the ventral cochlear nucleus

Pg 3, Ln 40 – please identify SBC

Pg 6, Ln 115 – a 10 current steps protocol should state the range of injected currents.

6. PLOS authors have the option to publish the peer review history of their article (what does this mean?). If published, this will include your full peer review and any attached files.

Reviewer #1: No

Reviewer #2: No

---

## [Author Response · Author response to Decision Letter 0]

14 Oct 2019

Detailed responses to comments: see coverletter of revision

---

## [Decision Letter · Decision Letter 1]

29 Oct 2019

PONE-D-19-18119R1

Muscarinic modulation of M and h currents in gerbil spherical bushy cells

PLOS ONE

Dear Dr Kuenzel

Thank you for submitting your manuscript to PLOS ONE. After careful consideration, we feel that it has merit but does not fully meet PLOS ONE’s publication criteria as it currently stands. Therefore, we invite you to submit a revised version of the manuscript that addresses the points raised during the review process.

Although the reviewers found the paper sufficiently improved, both of the have minor comments that need to be addressed to make the paper more readable. Please edit the manuscript following reviewers' suggestions.

We would appreciate receiving your revised manuscript by Dec 13 2019 11:59PM. To enhance the reproducibility of your results, we recommend that if applicable you deposit your laboratory protocols in protocols.io, where a protocol can be assigned its own identifier (DOI) such that it can be cited independently in the future. For instructions see: http://journals.plos.org/plosone/s/submission-guidelines#loc-laboratory-protocols

We look forward to receiving your revised manuscript.

Kind regards,

Andrea Barbuti, PhD

Academic Editor

PLOS ONE

Reviewers' comments:

Reviewer's Responses to Questions

**Comments to the Author**

1. If the authors have adequately addressed your comments raised in a previous round of review and you feel that this manuscript is now acceptable for publication, you may indicate that here to bypass the “Comments to the Author” section, enter your conflict of interest statement in the “Confidential to Editor” section, and submit your "Accept" recommendation.

Reviewer #1: All comments have been addressed

Reviewer #2: (No Response)

2. Is the manuscript technically sound, and do the data support the conclusions?

Reviewer #1: Yes

Reviewer #2: Partly

3. Has the statistical analysis been performed appropriately and rigorously? 

Reviewer #1: Yes

Reviewer #2: I Don't Know

4. Have the authors made all data underlying the findings in their manuscript fully available?

Reviewer #1: Yes

Reviewer #2: Yes

5. Is the manuscript presented in an intelligible fashion and written in standard English?

Reviewer #1: Yes

Reviewer #2: No

6. Review Comments to the Author

Reviewer #1: The authors have addressed all my concerns raised during the previous review. The new analyses and extended discussion further strengthen the authors’ conclusions and the current manuscript is much improved.

Consequently, I have only a few minor points I would like to point out:

- Line 334: “OxoM” should probably be lowercase.

- Line 364: “washing” should perhaps read “wash-in”

- Figure 7C: The 200Hz condition seems to show the mean values only with individual cells missing.

- Figure 10A: The Figure legend (Line 600f) describes the right panel as “after 10 min of oxoM perfusion” but the figure panel is labeled “recovery” (i.e. some time after drug application has been stopped)

- Figure 10A: Please label the middle panel “oxoM wash-in”, since multiple drugs (carbachol, oxoM, 4DAMP) have been used in the figure.

Reviewer #2: Although the authors corrected many grammatical errors, I suggest a further editing and proof reading to aid clarity and make the text more readable. Moreover the affiliation of the manuscript was a bit awkward as the figure legends were mixed into the text, without the figures, which to my opinion just adds to the confusion. Additional concerns:

In the methods section, the authors state that they chose to include cells with quite high Rs (up to 35 MOhm) and allowed fluctuations of 30%. All these parameters can significantly affect their recordings and therefore conclusions. As the authors state that no more experiments can be made, I suggest commenting on this in the discussion.

Ln 138. Please rephrase to “One way ANOVA with repeated measures”, as well as other places in the text.

Results section, ln 286 onwards: this section is written like a figure legend, which is quite repetitive, why?

Results in Fig 3A are a bit odd. The plot shows that there is almost no difference between the data points at -70mV, yet its deemed significant as there is a small Im current at this voltage?

Fig. 3D – maybe I’m missing something, but in the legend the authors state that the comparison was between the treatment to control conditions (no treatment), yet in the plot it seems like the authors compared between the different treatments. Please clarify. Also, does the results with OXOM+XE991 means that there was no inhibition of Im, or that compared to XE991 alone, there was no additional inhibition? This needs clarification!

The description of the measured currents (Im, Ih) repeat itself too many times ( methods, results and legend).

Ln. 385 – the conclusion that “ The presence of ZD7288 before oxoM application prevented large effects of oxoM on Ih” does not make any sense, as in the presence of ZD7288, there is no Ih, so where this expectation came from…

Ln 409: the overall conclusion is wrong. Essentially, all you show is that cholinergic agonist decrease the Ih, there is no evidence to support the conclusion that Ih play a role in the cholinergic modulation of SBC’s.

Fig 10 legend: the plots in A are not 10 current steps (as only two traces are shown). Moreover the labels in the plot suggest “wash-in” and “recovery”, while the legend suggest it is 3 min and 10 min post carbachol application. I think there is a mix-up with fig 9A, Please clarify.

I found result 10J a bit puzzling, and essentially contradict Fig 4f. So did OXOM decrease Ih or not?

7. PLOS authors have the option to publish the peer review history of their article (what does this mean?). If published, this will include your full peer review and any attached files.

Reviewer #1: No

Reviewer #2: No

---

## [Author Response · Author response to Decision Letter 1]

5 Dec 2019

Detailed responses to comments: see coverletter of revision.

---

## [Editor Report · Decision Letter 2]

11 Dec 2019

Muscarinic modulation of M and h currents in gerbil spherical bushy cells

PONE-D-19-18119R2

Dear Dr. Kuenzel,

We are pleased to inform you that your manuscript has been judged scientifically suitable for publication and will be formally accepted for publication once it complies with all outstanding technical requirements.

With kind regards,

Andrea Barbuti, PhD

Academic Editor

PLOS ONE
---

## [Editor Report · Acceptance letter]

19 Dec 2019

PONE-D-19-18119R2 

Muscarinic modulation of M and h currents in gerbil spherical bushy cells 

Dear Dr. Kuenzel:

I am pleased to inform you that your manuscript has been deemed suitable for publication in PLOS ONE. Congratulations! Your manuscript is now with our production department. 

With kind regards,

on behalf of

Dr. Andrea Barbuti 

Academic Editor

PLOS ONE